# Omics-Mediated Treatment for Advanced Prostate Cancer: Moving Towards Precision Oncology

**DOI:** 10.3390/ijms26157475

**Published:** 2025-08-02

**Authors:** Yasra Fatima, Kirubel Nigusu Jobre, Enrique Gomez-Gomez, Bartosz Małkiewicz, Antonia Vlahou, Marika Mokou, Harald Mischak, Maria Frantzi, Vera Jankowski

**Affiliations:** 1Department of Biomarker Research, Mosaiques Diagnostics GmbH, 30659 Hannover, Germany; fatima@mosaiques-diagnostics.com (Y.F.); jobre@mosaiques-diagnostics.com (K.N.J.); mokou@mosaiques-diagnostics.com (M.M.); mischak@mosaiques-diagnostics.com (H.M.); 2Institute for Molecular Cardiovascular Research (IMCAR), RWTH Aachen University, 52074 Aachen, Germany; vjankowski@ukaachen.de; 3Urology Department, Reina Sofía University Hospital, Maimonides Institute of Biomedical Research of Cordoba (IMIBIC), University of Cordoba (UCO), 14004 Cordoba, Spain; enrique.gomez.gomez.sspa@juntadeandalucia.es; 4Department of Minimally Invasive and Robotic Urology, Centre of Excellence in Urology, Wroclaw Medical University, 213 Borowska Street, 50-556 Wroclaw, Poland; bartosz.malkiewicz@umw.edu.pl; 5Systems Biology Center, Biomedical Research Foundation, Academy of Athens, 115 27 Athens, Greece; vlahoua@bioacademy.gr

**Keywords:** biomarker-guided therapy, combination therapy, multi-omics, personalized medicine, precision oncology, therapeutic interventions, treatment response prediction

## Abstract

Prostate cancer accounts for approximately 1.5 million new diagnoses and 400,000 deaths every year worldwide, and demographic projections indicate a near-doubling of both figures by 2040. Despite existing treatments, 10–20% of patients eventually progress to metastatic castration-resistant disease (mCRPC). The median overall survival (OS) after progression to mCPRC drops to 24 months, and efficacy drops severely after each additional line of treatment. Omics platforms have reached advanced levels and enable the acquisition of high-resolution large datasets that can provide insights into the molecular mechanisms underlying PCa pathology. Genomics, especially DDR (DNA damage response) gene alterations, detected via tissue and/or circulating tumor DNA, efficiently guides therapy in advanced prostate cancer. Given recent developments, we have performed a comprehensive literature search to cover recent research and clinical trial reports (over the last five years) that integrate omics along three converging trajectories in therapeutic development: (i) predicting response to approved agents with demonstrated survival benefits, (ii) stratifying patients to receive therapies in clinical trials, (iii) guiding drug development as part of drug repurposing frameworks. Collectively, this review is intended to serve as a comprehensive resource of recent advancements in omics-guided therapies for advanced prostate cancer, a clinical setting with existing clinical needs and poor outcomes.

## 1. Introduction

Prostate cancer (PCa) presents a significant global health challenge, ranking as the second most prevalent cancer among men worldwide and a major contributor to cancer-related deaths [1]. The significant impact of PCa is highlighted by over 1.4 million new diagnoses and nearly 400,000 associated deaths reported globally in 2020 [1], with estimates for 2023 in the United States predicting 288,300 new cases and 34,700 deaths [2]. Although localized PCa generally has a favorable outcome, progression to advanced and metastatic stages dramatically decreases the 5-year survival rate to between 29–30% [2,3]. Particularly concerning is metastatic castration-resistant prostate cancer (mCRPC), which develops after acquiring resistance to androgen deprivation therapy (ADT) [4]. Heterogeneity is observable both between patients and within individual tumors [5,6]. This inherent biological variability makes therapeutic interventions more challenging since standard treatments like ADT, chemotherapy, and radiation therapy often face significant limitations [7,8]. The lack of accurate predictive, stratification, and/or prognostic tools can lead to suboptimal management of advanced, aggressive disease [9,10]. Thus, a critical and immediate clinical need does exist for tailored management approaches to guide treatment for patients with advanced PCa [11].

Current clinical practices often depend on traditional imaging modalities, including bone scans, computed tomography (CT), and magnetic resonance imaging (MRI), for monitoring disease progression and metastasis [11]. Nevertheless, these approaches do not yield crucial molecular insights regarding the underlying tumor pathophysiology [12]. Emerging technologies, such as prostate-specific membrane antigen positron emission tomography-computed tomography (PSMA PET-CT), have shown improved sensitivity in detecting micro-metastases [13,14]. However, their effectiveness in tailoring personalized treatments is limited, particularly for neuroendocrine or small-cell variants that show low or no PSMA expression [15]. Additionally, established clinical markers like prostate-specific antigen (PSA) levels prove insufficient for precise prognostic assessments, or for determining the right timing for treatments, as they cannot distinguish between slow-growing and aggressive forms, or forecast individual patient reactions to therapies [16].

The rise of “omics” technologies, covering genomics, transcriptomics, proteomics, and metabolomics, has led to a new age of personalized medicine concerning the diagnosis, prognosis, and treatment of Pca [17,18]. These technologies allow for the concurrent evaluation of many molecular characteristics, offering a deep understanding of tumor molecular profiles and uncovering the detailed biological mechanisms underlying disease pathology [19]. A significant advantage of omics is its ability to unveil tumor heterogeneity, which is reflected in the considerable variability both between patients and within tumors themselves, complicating the implementation of broad diagnostic and treatment strategies [20]. By analyzing the genomic, transcriptomic, and proteomic profiles of specific tumors, omics enabled the discovery of new drug targets and resistance mechanisms; for instance, mutations in DNA damage repair (DDR) genes like *BRCA1/2* and *ATM* can indicate sensitivity to PARP inhibitors, while alterations in the androgen receptor (AR) pathway may inform AR-targeted therapies [11,21]. Additionally, omics has provided valuable predictive and prognostic insights by allowing for the stratification of patients to select the most effective therapies and monitor disease progression. Omics-derived tests based on gene expression profiles, such as Decipher [22], Oncotype DX [23], and Prolaris [24,25], improve risk assessment for PCa progression and monitoring beyond traditional clinical variables [26]. The combination of multi-omics data holds great promise for discovering biomarkers and translating these findings into clinical practice in the management of PCa [27]. This integrated strategy, along with advanced methods such as liquid biopsies and “ex vivo” patient-derived organoids, enables real-time tracking of tumor changes and modifications to therapy, thus providing a more dynamic perspective on the disease [28].

In today’s landscape of precision oncology, the treatment approach for advanced PCa is experiencing a significant transition from broad to targeted individual or in combination therapies driven by molecular insights [29]. This change is driven by a better understanding of the detailed molecular architecture of tumors and their interaction with the microenvironment, which aids in pinpointing essential molecular pathways and biomarkers that inform personalized treatment choices [29]. A key example of a precision medicine approach is the approval of PARP inhibitors (such as olaparib, rucaparib, niraparib, and talazoparib) for mCRPC guided by a companion test, based on DDR gene alterations, including mutations in *BRCA1/2* and *ATM* [30,31]. The success of studies like PROfound, which showed PARP inhibition superiority to traditional AR-targeted treatments in biomarker-selected patient groups, highlights the vital importance of genomic stratification within clinical practice [32,33,34]. Likewise, AR-targeted therapies, comprising effective AR pathway inhibitors like abiraterone and enzalutamide, continue to serve as essential treatments [35]. Omics-based monitoring, especially through liquid biopsies, allows for real-time observation of AR splice variants (such as *AR-V7*) that indicate resistance, facilitating timely treatment modifications [36]. This approach of selecting patients based on biomarkers is significantly improving the design of clinical trials and therapeutic strategies, necessitating thorough genomic characterization, which includes both somatic and germline analyses, to match patients with the most effective treatments and enhance sequential therapeutic plans [37]. Furthermore, rather than relying solely on single-agent strategies, insights from omics are progressively driving the creation of combination therapies designed to characterize treatment resistance and improve patient outcomes by investigating synergistic combinations, such as AR inhibition paired with DDR-targeted therapies or immunotherapy approaches to address tumor adaptability and the immunosuppressive microenvironment present within tumors [38]. Figure 1 shows the areas of applications for omics-mediated interventions in PCa.

Distinct molecular profiles of PCa patients highlight that no single therapy can be universally applied [39]. In this context, omics technologies, including genomics, transcriptomics, proteomics, metabolomics, and innovative liquid biopsy techniques, have facilitated a better characterization of PCa at the molecular level, by allowing for simultaneous analysis of thousands of molecular characteristics [29]. These innovations have been crucial in uncovering new biomarkers, such as particular gene modifications (e.g., in *AR*, *TP53*, *PTEN*, *BRCA1/2*, *ATM*, *CDK12*, *ETS* family, *SPOP*, *FOXA1*), RNA alterations, circulating tumor DNA (ctDNA), and microRNAs, which together inform prognosis and assist in treatment selection [11,40]. Additionally, omics-driven research has pinpointed and confirmed potential drug targets, leading to therapies such as PARP inhibitors for tumors deficient in DDR, agents targeting AR, inhibitors of the PI3K/AKT/mTOR pathway, and PSMA-targeted radioligand therapies as shown in Figure 2. This review seeks to consolidate the present status of these omics-based treatment strategies for advanced PCa, emphasizing their essential role in precision oncology.

## 2. Methods

We conducted a PRISMA-guided literature search in the Web of Science Core Collection (chosen for its broad coverage of translational and clinical oncology) to identify pharmacological interventions for advanced-stage PCa supported by omics derived signatures or network analyses. Three tailored Boolean strategies (with detailed description on search queries provided in Appendix B) were run for the unified window between 1 January 2021 and 30 April 2025, as following: (i) approved therapies paired with a predictive/companion biomarker or biomarker assay, (ii) investigational studies in late drug developmental phase (phase II/III) clinical trials in which patients are stratified based on omics-based markers, and (iii) repurposed drug candidates proposed through multi-omics or network-based approaches.

The comprehensive search strategy incorporated relevant keywords, synonyms, and appropriate search filters to ensure a robust identification of relevant studies. Only studies published in English were considered. Conference abstracts, articles in press, books, book chapters, commentaries, methodological papers, review articles, editorials, and non-human studies were excluded. Two authors independently screened each article based on the title and abstract. Any discrepancies in inclusion decisions were resolved through consultation with a third author. The screening was facilitated using the web-based software tool Rayyan [41]. Figure 3a–c show the PRISMA flow diagrams detailing the study selection process for predictive/companion omics-based approved therapies, omics-based stratification tools for investigational treatments in clinical trials, and repurposed drug candidates proposed through multi-omics or network-based approaches, respectively. Appendix A list all articles that were included in this review, for the first, second, and third searches, respectively. Appendix A provide a comprehensive list of excluded articles with corresponding reasons for exclusion.

Exclusion criteria included: (i) publication type (review articles, editorials, commentaries, etc.), (ii) irrelevant study outcomes (that were limited to observational biomarker analyses, purely mechanistic investigations without testing a therapeutic intervention), (iii) study population (studies not performed on PCa patients and all non-human studies), (iv) study design (excluded clinical studies with fewer than 50 patients in the omics-characterized cohort), (v) articles written at foreign languages.

Study characteristics and outcome data were systematically collected using a standardized extraction form. The gathered variables included, but were not limited to: the study title, sample size, participant demographics, inclusion and exclusion criteria, length of the intervention and follow-up period, study design, details of the intervention and comparator or control, primary and secondary outcomes, any reported adverse events or complications, key findings, effect sizes with statistical significance, as well as the authors’ conclusions, noted strengths, and limitations of the study.

## 3. Results

### 3.1. Omics-Based Predictive Markers to Guide Approved Interventions in Advanced PCa

Advanced PCa, encompassing both metastatic hormone-sensitive (mHSPC) and mCPRC, has historically been managed with broadly acting hormonal agents and chemotherapy [42]. In the past five years, however, regulatory approvals have increasingly adopted the integration of molecular assays, ranging from next-generation sequencing (NGS) and transcriptomic classifiers to PET-based imaging biomarkers to identify the subsets of patients most likely to benefit from the given therapeutic regimes. Under the FDA’s drug-diagnostic co-development paradigm, each new systemic agent approved for advanced PCa is paired with a companion diagnostic or biomarker assay, adopting a more precision-medicine framework [43]. By using omics technologies to predict which patients are a good fit for receiving targeted therapy based on DNA repair deficiency, tumor antigen expression, or mutational burden, these interventions move beyond “one-size-fits-all” paradigm to deliver targeted therapies only to molecularly defined subpopulations thereby maximizing efficacy, minimizing unnecessary toxicity, and setting the stage for future label expansions driven by emerging biomarker insights.

#### 3.1.1. PARP Inhibitors

A pivotal breakthrough for mCPRC has been the development of poly (ADP-ribose) polymerase (PARP) inhibitors for tumors harboring defects in the homologous-recombination repair (HRR) pathway. Mutations in these genes are recognized as critical biomarkers in mCRPC. Pharmacologic blockade of PARP impairs the single-strand break repair mechanism, creating synthetic lethality that selectively eliminates HRR-deficient cancer cells while sparing HRR-proficient tissue [44]. For treatment initiation, the FDA mandates a certified NGS assay such as FoundationOne CDx, BRACAnalysis CDx for tissue, or FoundationOne Liquid CDx for plasma to identify any HRR gene alterations [45]. A multi-center analysis including 123 mCRPC patients treated with olaparib, rucaparib or talazoparib (all PARP inhibitors) revealed marked gene-specific heterogeneity: *BRCA2*-mutant tumors achieved a median radiographic progression free survival (rPFS) of 8.2 months, compared to 3.5 months in *BRCA1*-mutant disease, respectively [Hazard ratio (HR) = 2.08, 95% CI, 0.99–4.40; *p* < 0.05)] [46]. Along these lines, in a cohort of 9292 advanced tumors clinical evidence demonstrated that *BRCA1/2* or broader HRR-gene mutations alone were not independent prognosticators for disease outcome, whereas a genome-wide homologous recombination deficiency (HRD) scar signature was associated with a 22% reduction in overall survival (OS), implying that the HRD genome-wide assay could prioritize earlier PARP inhibitor treatment [47]. These findings suggest that on the one hand, comprehensive HRD screening tests may outperform single-gene panels, and on the other, that not all *BRCA* alterations confer equal PARP inhibitor sensitivity, principles that increasingly inform trial design and clinical decision-making.

In particular, olaparib’s FDA approval was pursued based on the PROfound clinical trial results (NCT02987543). Eligible patients included those with mCRPC already progressed after enzalutamide or abiraterone treatment and with confirmed presence of HRR gene alterations. In the arm where patients had *BRCA1/2* or *ATM* mutations, olaparib (300 mg twice daily) more than doubled the median rPFS time compared to the physician’s choice of abiraterone/prednisone or enzalutamide, resulting in 7.4 months versus 3.6 months (HR = 0.34; 95% CI, 0.25–0.47; *p* < 0.001). The median OS was additionally extended to 19.1 months versus 14.7 months in the control arm, where no mutations were considered (HR = 0.69; 95% CI, 0.50–0.97; *p* = 0.02) [44,48]. Enrollment required confirmation of HRR alterations via FoundationOne CDx or BRACAnalysis CDx on tumor tissue. Along these lines, in a dedicated real-world clinical setting study, 4858 formalin-fixed paraffin-embedded samples from 4047 men were screened with NGS, resulting in the successful assessment of only 58% of specimens (69% of patients). Failure in NGS screening was driven largely by the low tumor content and the quality of archival tissue [49]. A prespecified sub-study subsequently showed that high-quality plasma NGS (FoundationOne Liquid CDx/GuardantOMNI) detected *BRCA*/*ATM* alterations in 80% of initially tested patients and reproduced the treatment effect (ctDNA-positive rPFS 7.4 vs. 3.5 months; HR = 0.33; 95% CI, 0.21–0.53), leading to FDA clearance of liquid biopsy as an alternative companion diagnostic [50]. The most recent post hoc analysis of PROfound results focusing on the 160 patients with *BRCA1/2*-altered tumors confirmed durable benefit irrespective of germline or somatic origin of the mutations (rPFS 9.8 vs. 3.0 months; HR = 0.22; OS 20.1 vs. 14.4 months; HR = 0.63) and revealed particularly longer survival benefit in cases with *BRCA2* homozygous deletions (median rPFS 16.6 months) [51]. Finally, a retrospective real-world analysis of 445 mCRPC patients in the Flatiron Health–Foundation Medicine clinico-genomic database treated with single-agent PARP inhibitors showed that men with homozygous *BRCA1/2* deletions derived the most durable benefit, with median time-to-next-treatment 19.4 vs. 9.0 months and OS of 19.4 vs. 14.7 months compared with other pathogenic *BRCA* variants, highlighting the need for tissue NGS or high-tumor-fraction (≥20%) ctDNA assays to detect these deep deletions [52].

Similarly, rucaparib received accelerated FDA approval in May of 2020 based on the strength of the single-arm, phase II TRITON2 trial (NCT02952534) [53]. Eligible men include those with diagnosed mCRPC that had progressed after both a next-generation androgen-receptor pathway inhibitor and a taxane, who harbored germline or somatic *BRCA1/2* alterations, confirmed by testing of tumor tissue or plasma sequencing (in this case, ctDNA) in a central reference laboratory, or through a validated local assay. Among others, central testing revealed only 47% concordance between paired plasma and tissue samples in patients with somatic *BRCA* alterations, underscoring both the limitations of liquid biopsy and the persistent need for robust companion-diagnostic algorithms [54]. Among the 62 patients bearing *BRCA* mutations who met the trial entry criteria and received rucaparib (600 mg twice daily), the confirmed objective response rate was 44% (95% CI, 31–57%), with 56% of responders maintaining benefit for over 6 months [55]. These data translated into an accelerated approval that now awaits verification based on the randomised TRITON3 trial. In TRITON3, rucaparib significantly prolonged imaging-based progression-free survival in comparison to physician’s choice between docetaxel, abiraterone/prednisone, or enzalutamide, for 11.2 months versus 6.4 months in the *BRCA* subgroup (HR = 0.50; 95% CI, 0.36–0.69; *p* < 0.001) [56].

While genomics testing for the presence of HRR mutations is a prerequisite for receiving PARP inhibitors, as evident based on the above clinical trial data, several technical limitations exist. A high percentage (approximately 25–35%) of patients fail the screening, due to sub-optimal (for genomics testing) archival tissue quality and/or disconcordance of mutation status between peripheral plasma and tissue (up to 80% depending on the targeted mutations) [57,58].

#### 3.1.2. Combinatorial Therapies

Application of different drugs in combination is also gaining popularity based on respective clinical trial results. Olaparib gained U.S. FDA approval on 31 May 2023 for administration together with abiraterone acetate (1000 mg once daily) and prednisone/prednisolone (5 mg twice daily) in adults with deleterious or suspected-deleterious *BRCA*-mutated mCRPC [59,60]. The decision was based on the double-blind, phase III PROpel trial (NCT03732820), which randomised 796 chemotherapy-naïve men in an one-to-one randomisation to receive olaparib (300 mg twice daily) with abiraterone/prednisone (*n* = 399), or those receiving a matched placebo in combination with abiraterone/prednisone (*n* = 397) [60]. Median rPFS improved from 16.6 months with abiraterone alone to 24.8 months under the combinatorial therapy (HR = 0.66, 95% CI 0.54–0.81; *p* < 0.0001). An exploratory analysis of the 85 patients bearing *BRCA*-mutations showed an even greater benefit: median rPFS was not reached compared to 8 months for placebo (HR = 0.24, 95% CI 0.12–0.45) [59]. Although within the PROpel trial, patients were enrolled irrespective of HRR status, the FDA ultimately approved the above therapeutic scheme to *BRCA*-mutant disease after concluding that efficacy in non-*BRCA* tumors did not outweigh added toxicity. Consequently, molecular profiling is now recommended through any of the FDA-cleared companion diagnostics such as FoundationOne CDx or BRACAnalysis CDx [60]. A major consideration within combinatorial therapies is safety, as combination therapies are frequently related to increased side effects. Considering the combination of olaparib and abiraterone acetate, safety was class-consistent (i.e., presenting with equal side effects to other PARP inhibitors). Severe (Grade ≥ 3) anaemia occurred in 15–16% of olaparib-treated patients vs. approximately 3% with placebo; 18% required at least one red-cell transfusion. Other common grade ≥ 3 toxicities included hypertension (approximately 4%) and urinary-tract infection (approximately 2%). Treatment-related adverse events led to permanent discontinuation in approximately 14% of patients on the combination, supporting a manageable, predominantly haematologic toxicity profile when regular blood counts and dose modifications are employed [61]. Similarly, niraparib combined with abiraterone acetate/prednisone was investigated in a randomised, double-blinded, placebo-controlled phase III trial called MAGNITUDE (NCT03748641), which enrolled 423 chemotherapy-naïve men with mCRPC, all bearing HRR gene alterations [62]. Central (in a designated reference laboratory)next-generation sequencing of tumor tissue and/or plasma identified 225 (53%) patients with *BRCA1/2* mutations. Patients were subsequently randomised in a one-to-one setting to receive niraparib (200 mg once daily) plus abiraterone (1000 mg once daily) and prednisone (10 mg daily; *n* = 212), or matched placebo plus abiraterone and prednisone (*n* = 211). Within 18.6 months median follow-up, niraparib in combination with abiraterone acetate/prednisone more than halved the risk of radiographic progression or death in the group of patients bearing *BRCA* mutations (median rPFS 16.6 vs. 10.9 months; HR = 0.53, 95% CI 0.36–0.79; *p* = 0.0014) and prolonged rPFS in patients with HRR mutations (16.5 vs. 13.7 months; HR = 0.73, 95% CI 0.56–0.96; *p* = 0.0217) [62]. OS analysis (median follow-up 37.3 months) showed no statistically significant benefit in either group of patients with HRR mutations (HR = 0.93, 95% CI 0.72–1.20) or those with *BRCA* mutations (HR = 0.79, 95% CI 0.55–1.12). As in the case of olaparib, in this combination, major adverse effects were grade ≥ 3 anaemia occurring in 29.7% of niraparib-treated patients versus 7.6% with placebo, followed by thrombocytopenia and hypertension the next most common high-grade events; treatment-related adverse events led to permanent discontinuation in 15% of patients receiving niraparib [62]. Similarly, talazoparib (0.5 mg once daily) combined with enzalutamide (160 mg once daily) earned its regulatory footing on the randomized, placebo-controlled phase III TALAPRO-2 study (NCT03395197) [63]. Among 805 men with untreated mCRPC, 402 received the doublet and 403 received placebo and enzalutamide; prospective tumor testing identified HRR alterations in 399 of them across two sequential cohorts. In the intention-to-treat population, talazoparib more than halved the risk of radiographic progression or death median rPFS was not reached versus 21.9 months with enzalutamide alone (HR = 0.63, 95% CI 0.51–0.78; *p* < 0.001) after approx. 24 months’ follow-up. The benefit was even more pronounced in the HRR-deficient subgroup, where median rPFS again remained unreached compared with 13.8 months for control (HR = 0.45, 95% CI 0.33–0.61; *p* < 0.0001) [63]. Recently updated TALAPRO-2 data presented at ASCO-GU 2025 showed a significant OS’ advantage that persisted despite immature censoring, confirming a durable clinical benefit [64]. Patient enrollment mandated central NGS of tumor tissue (FoundationOne CDx) to screen for HRR mutations, based on which roughly 20% of screened samples lacked adequate material or qualifying mutations [64]. Among the 805 men who met these molecular criteria and were subsequently randomised, talazoparib plus enzalutamide produced a clinically meaningful improvement in radiographic progression-free survival versus enzalutamide alone, with the largest benefit observed in the HRR-mutated subgroup, thereby positioning the combination as a promising first-line therapy for mCRPC [64]. The overview of the findings and important points for PARP inhibitors are given in Table 1.

As in the targeted monotherapies, also in the combinatorial setting, routine multi-gene NGS of both tissue and circulating tumor DNA is now the standard of care, with the 2025 NCCN guideline mandating HRR-gene screening (BRCA1/2, ATM, PALB2, etc.) for every mCRPC patient [65]. Clinical applicability is direct, as the FDA’s 2023 approval of the combination of olaparib plus abiraterone/prednisone for BRCA-mutated mCRPC patients is tied to a companion diagnostic test such as, FoundationOne CDx and FoundationOne Liquid CDx, that can be run on archival tissue, or plasma [66]. Similarly, in 2024, the fixed-dose niraparib/abiraterone tablet (Akeega) was approved together with NGS screening with FoundationOne Liquid CDx as a companion test. Yet, despite these landmark advances, several limitations and uncertainties complicate the integration of PARP inhibitors into the routine management of mCRPC. First of all, conflicting clinical findings highlight that not all HRR gene mutations confer equal sensitivity, with *BRCA2*-mutant tumors achieving significantly better outcomes than *BRCA1* or other HRR-altered subtypes [67]. NGS screening approaches also come with technical limitations as tissue-based NGS screening frequently fails due to inadequate tumor material, and plasma-based assays show only modest concordance with the tissue mutational status. Additionally, combination therapies with PARP inhibitors (PARPi), especially with chemotherapy or other DNA-damaging agents, result in increased toxicity, mainly hematologic effects (anemia, neutropenia, thrombocytopenia) which often exceed those seen with monotherapy [68].

#### 3.1.3. Imaging-Based PSMA (Prostate-Specific Membrane Antigen) Directed Radioligand Therapy

PSMA is a transmembrane folate hydrolase enzyme that was found with increased protein expression levels in PCa [69]. As such, this protein has served as a cell-surface imaging biomarker and therapeutic target in advanced PCa. Nevertheless, as PSMA expression is often suppressed in AR-negative advanced prostate cancers and liver metastases, a definition of PSMA positivity is a prerequisite for the use of PSMA-radioligand therapy; in addition, PSMA heterogeneity across tumoral and spatial, inter-patient, intra-patient, lineage, and temporal dimensions presents a challenge for efficient patient management [69,70].

Historically, the identification of PSMA as a key proteomic biomarker has led to the development and approval of PSMA-targeted radioligand therapy, representing a significant advancement in the theranostic approach to advanced PCa. The phase III VISION trial showed that adding ^177Lu-PSMA-617 to the best standard of care in 831 men with PSMA-positive mCRPC prolonged OS from 11.3 to 15.3 months (HR = 0.62, *p* < 0.001) and improved radiographic progression-free survival, prompting US-FDA approval of lutetium-177 vipivotide tetraxetan (Pluvicto™) on 23 March 2022, together with same-day clearance of the companion imaging agent gallium-68 gozetotide (^68Ga-PSMA-11, Locametz) that must verify PSMA-positive disease by PET/CT before therapy [71]. Another post hoc VISION analysis (Table 2) revealed a graded association between early (≤12-week) PSA decline and later radiographic progression-free survival, overall survival, and quality-of-life outcomes. Notably, VISION excluded patients who had any PSMA-negative lesion measuring ≥2.5 cm in lymph nodes or ≥1 cm in visceral or bone sites, meaning the OS benefit cannot be extrapolated to tumors with mixed or low PSMA expression [71].

Building on these late-line data, the phase III PSMAfore trial randomised 468 taxane-naïve patients who had progressed on a single androgen-receptor-pathway inhibitor (ARPI) to six cycles of ^177Lu-PSMA-617 or a switch to the alternate ARPI. At the third data cut (median follow-up 24 months), ^177Lu-PSMA-617 halved the risk of radiographic progression or death (median rPFS 11.6 vs. 5.6 months; HR = 0.49, 95% CI 0.39–0.61) while producing fewer grade 3–5 adverse events (36% vs. 48%), despite 57% cross-over from the control arm [72]. Despite the impressive rPFS hazard ratio, 8% of screened patients were ineligible due to discordant PSMA uptake, underscoring the need for back-up imaging in heterogeneous disease. OS data remain immature (crossover-adjusted HR = 0.80; HR 0.41; *p* < 0.0001), yet the consistency of efficacy and safety across both VISION and PSMAfore supports the integration of PSMA-targeted radioligand therapy not only after but also before taxane chemotherapy, further consolidating PSMA as a central biomarker and therapeutic target in advanced prostate cancer [72].

**Table 2 ijms-26-07475-t002:** FDA-approved PSMA-directed radioligand therapy for PSMA-positive mCPRC.

Combination/Dose	Molecular Requirement (Companion Diagnostic)	Pivotal Trial (s)	Key Treated Population (*n*)	Key Efficacy Outcome	*p* Value (Primary End-Point)	Notable Points
Lutetium-177 vipivotide tetraxetan (Pluvicto™; ^177Lu-PSMA-617) 7.4 GBq i.v. every 6 weeks × 6 cycles and best standard care vs. care alone [71]	PSMA-positive on ^68Ga-PSMA-11 PET/CT (≥1 lesion hotter than liver; no PSMA-negative lesion > 1 cm)	VISION (phase III)	Post-ARPI & taxane mCRPC; PSMA-PET-positive; *n* = 831 (551 vs. 280)	OS 15.3 vs. 11.3 mo (HR 0.62); rPFS 8.7 vs. 3.4 mo (HR 0.40)	OS *p* < 0.001; rPFS *p* < 0.001	- FDA cleared Pluvicto™ + Locametz™ on 23 Mar 2022. ≥50% PSA fall by week 12 → longer OS & better QoL. - Baseline whole-body TLP outperforms SUVmax for OS prediction DDR-gene alterations not predictive. - Grade ≥ 3 AEs: 52% vs. 38% (xerostomia, nausea, anaemia).
Lutetium-177 vipivotide tetraxetan (Pluvicto™; ^177Lu-PSMA-617) 7.4 GBq i.v. every 6 weeks × 6 cycles with best standard care vs. **change of ARPI** (abiraterone and enzalutamide alternative switching) [72]	PSMA-positive on ^68Ga-PSMA-11 PET/CT (≥1 lesion hotter than liver, no PSMA-negative lesion > 1 cm)	PSMAfore (phase III)	Post-ARPI, taxane-naïve mCRPC; PSMA-PET-positive; *n* = 468 (234 vs. 234)	Primary rPFS 9.30 vs. 5.55 mo—HR 0.41 (95% CI 0.29–0.56)	*p <* 0.0001 *(two-sided)*	- 57% of control pts crossed over to RLT, diluting OS signal. - Updated rPFS at 3rd cut: 11.6 vs. 5.6 mo—HR 0.49. - Interim OS: 23.66 vs. 23.85 mo—HR 0.98, *p* = 0.44. - Grade ≥ 3 AEs: 36% vs. 48% (lute-177 vs. ARPI). - Improved time to pain/QoL deterioration relative to ARPI switch.

Abbreviations: ^177Lu = lutetium-177; ^68Ga = gallium-68; AEs = adverse events; ARPI = androgen-receptor-pathway inhibitor; CT = computed tomography; DDR = DNA-damage repair; GBq = gigabecquerel; HR = hazard ratio; i.v. = intravenous; mCRPC = metastatic castration-resistant prostate cancer; mo = months; *n* = number of patients; OS = overall survival; PET = positron emission tomography; PSA = prostate-specific antigen; PSMA = prostate-specific membrane antigen; QoL = quality of life; rPFS = radiographic progression-free survival; SUVmax = maximum standardised uptake value; TLP = total-lesion PSMA).

In another multicentric cohort of 137 men, Rami et al. evaluated whether alterations in the DDR pathway were associated with outcomes to LuPSMA and reported that pathogenic DNA-damage-repair mutations (*BRCA1/2*, *ATM*, etc.) had no impact on PSA response, radiographic progression free, or OS after ^177Lu-PSMA-617, so DDR status should not be used to exclude patients [73]. Further retrospective analysis from the prospective REALITY Study (NCT04833517) in 102 mCRPC patients undergoing ^177Lu-PSMA-617 demonstrated that while the mean standardized uptake value (SUVmean) of all lesions on pre-therapeutic ^68Ga-PSMA-11 PET/CT was significantly associated with PSA response, only whole-body total-lesion PSMA (TLP) emerged as an independent predictor of OS in multivariable analysis (HR = 1.033, *p* = 0.009). This finding, highlighting TLP’s superior performance over other tested parameters like SUVmax for survival prediction, further underscores the crucial value of comprehensive volumetric tumor-burden metrics in prognostication for PSMA RLT [74]. Together, these data endorse PSMA-PET positivity as the mandatory companion test while highlighting volumetric PET metrics and on-treatment PSA kinetics as promising adjuncts.

Importantly, as PSMA expression is suppressed by AR, AR-positive tumors typically exhibit low PSMA expressions. Upon AR pathway inhibition, there may be a consequent up-regulation of PSMA; however, if the tumor undergoes neuroendocrine differentiation (NED), driven by factors like N-Myc and EZH2, epigenetic silencing of FOLH1 results in PSMA-low or -negative subclones [75]. At the same time, lineage plasticity is the ability of tumor cells to shift between luminal, neuroendocrine, and other cell phenotypes. It not only enables survival under therapeutic pressure but also fosters the emergence of treatment-resistant phenotypes characterized by low or absent PSMA expression [76]. PSMA negativity then predicts poor response to ^177Lu-PSMA-617 radioligand therapy. Transient “PSMA flare” after brief AR inhibition shortly after starting AR inhibition therapy (ADT or AR blockers) can increase PSMA target density and may be exploited to enhance the efficacy of PSMA-directed therapies. Short-term androgen blockade (termed STAB) has been shown to transiently upregulate PSMA expression in PCa, resulting in SUV_max_ increases of approximately 30–50% on PSMA PET imaging. Integration of STAB into imaging or treatment protocols may improve diagnostic sensitivity and therapeutic efficacy, particularly in cases where baseline PSMA expression is suboptimal. Nevertheless, further prospective studies are warranted to standardize timing and patient selection for this sensitization approach [77]. Recognizing dynamic changes in PSMA expression is clinically critical for imaging interpretation, therapy selection, and realizing the potential of combination or sequential therapies.

The clinical detectability and quantification of PSMA expressions in PCa lesions are directly influenced by the choice of radiotracer and imaging technology. Although [^68Ga]-labeled agents have traditionally been the clinical standard, next-generation fluorinated compounds like [^18F]DCFPyL offer advantages, including improved image resolution and delayed imaging protocols (referring to acquisition of image scans at 90–120 min, or even up to 3 h post-injection, rather than the standard image acquisition within an hour), which have demonstrated superior sensitivity for local and nodal staging in high- and very high-risk patients. For example, a recent study demonstrated that [^18F]DCFPyL PET/CT significantly improves the detection of loco-regional disease in patients with advanced PCa who are candidates for radical therapy, ensuring more accurate risk stratification, surgical planning, and more effective prioritization for PSMA-targeted therapies [78].

Moreover, in AR-negative double-negative prostate cancer (DNPC), PSMA expression can be virtually absent despite high metabolic activity, whereas AR-positive CRPC often retains heterogeneous PSMA uptake [75]. These resistant subpopulations, often AR-negative or double-negative, display marked molecular and metabolic divergence from classical adenocarcinoma, including upregulation of glucose transporters and hexokinases that facilitate increased glycolytic activity. Recent papers show PSMA-low tumors overexpress sugar transporters (GLUT1) and hexokinases, increasing the uptake in an FDG (glucose) PET scan, even when PSMA-PET signal is low. This metabolic shift provides a mechanistic rationale for integrating ^18F-FDG PET in the diagnostic workflow for PSMA-low or neuroendocrine tumors, complementing ^68Ga-PSMA PET [79]. In one case report, FDG-PET/CT identified lesions not seen on PSMA-PET/CT, underscoring the potential for FDG-positive and PSMA-negative discordance [80]. This means that while PSMA-PET/CT did not show any activity in these areas, FDG-PET/CT identified lesions suggestive of active disease. Excluding patients with FDG-positive but PSMA-negative disease from PSMA-targeted RLT (Radioligand therapy) improves response rates and spares them from an ineffective treatment, and should be triaged to alternative RLT targets such as GRPR or DLL3 [81,82,83].

Recent studies further highlight the promise of advanced biomarker approaches, such as radiomic and metabolomic profiling (discussed in Section 3.5 of this review), which leverage high-dimensional imaging and molecular data to capture clinically relevant disease heterogeneity that is not apparent with standard imaging alone. These mechanistic insights explain both the variable responses observed in PSMA-targeted radioligand trials.

### 3.2. Stratification Omics-Based Markers Used in Active Clinical Trials

#### 3.2.1. AKT/PI3K Pathway Trial for PTEN Deficient Disease

Loss of the tumor suppressor *PTEN* (phosphatase and tensin homolog), which occurs in 40–50% of advanced prostate cancers, leads to hyperactivation of the PI3K–AKT signaling pathway. Omics assays (genomic profiling) can identify *PTEN*-deficient tumors, which might respond to AKT inhibitors. A phase 3 global trial (IPATential150, NCT03072238) stratified patients by *PTEN* loss status. Treatment- naïve men diagnosed with mCRPC were randomized to receive either ipatasertib (AKT inhibitor) plus abiraterone/prednisone, compared to placebo plus abiraterone/prednisone [84]. The trial used two co-primary endpoints: investigator-assessed rPFS in the *PTEN* loss subgroup and rPFS in the overall (intention-to-treat) population. *PTEN* loss was detected by immunochistochemistry and confirmed with NGS. Results showed a significant benefit in rPFS in *PTEN* loss tumors with median time 18.5 vs. 16.5 months (HR = 0.77, *p* = 0.034) at an ipatasertib of 400 mg QD, in combination with abiraterone/prednisone). Strikingly, no benefit was shown in the unstratified population (HR = 0.84, *p* = 0.04; α-bound 0.01). As of 2025, at a follow-up time of approximately 34 months this did not translate into an overall-survival advantage (*PTEN* loss HR = 0.94; ITT HR = 0.91). As expected, based on the combination therapies, toxicity was higher in patients receiving ipatasertib (grade ≥ 3 events 70% vs. 39%), driven mainly by rash, diarrhoea, and hyperglycaemia. Ipatasertib added to abiraterone helps delay progression in *PTEN* null mCRPC but provides minimal benefit in unstratified via *PTEN* patients. This underscores that *PTEN* omics status can guide therapy: AKT inhibition is a promising strategy only in the molecular subgroup with pathway activation. Similarly, the IPATential150 trial demonstrated improved outcomes with ipatasertib in genomically-selected (*PTEN* null, protein expression lost) mCRPC. This benefit did not extend to unselected patients, highlighting the need for molecular stratification when using pathway-targeted agents [84].

#### 3.2.2. Epigenomics Assays to Stratify Patients Before Receiving Chemotherapy

GUIDE (ANZUP 1903; ACTRN12621001003742) is the first prospective trial to use a circulating epigenetic marker, which is plasma-methylated *GSTP1* (*mGSTP1*), to guide cytotoxic chemotherapy in mCRPC. In GUIDE, clearance of circulating *mGSTP1* is assessed centrally from 1 mL of plasma. As such, cell-free DNA undergoes bisulphite conversion followed by a quantitative head-loop methylation-specific PCR assay, while detectable *mGSTP1* is defined as >1 ng methylated *GSTP1* DNA per sample [85]. Men with detectable *mGSTP1* at baseline are randomised in a two-to-one setting to intermittent versus continuous 3-weekly docetaxel [85]. The primary end-point is investigator-assessed rPFS; secondary outcomes include time on treatment holidays, grade ≥ 3 toxicity, QoL, overall val, and cost-effectiveness. The protocol initially sought to enroll 120 patients across Australian centers, but slow accrual led to an amended target of 28 patients. According to the ANZUP 2024 annual report, only six men had been randomised by 31 March 2024, and the trial has since been listed as “Closed” on the ANZUP website, with no efficacy results having been released so far [85,86].

#### 3.2.3. Lipidomics-Based Stratification of mCRPC Patients to Receive a PCSK9 Inhibitor

Scheinberg et al. recently described the PCPro score, a five-analyte circulating lipidomic signature composed of three ceramides [Cer(d18:1/18:0), Cer(d18:1/24:0), Cer(d18:1/24:1)] plus total triglycerides and total cholesterol, which stratifies poor prognosis in mCRPC [87]. The three ceramides are quantified with a single-run, regulatory-compliant liquid chromatography tandem mass-spectrometry (LC-MS/MS) assay (positive-mode electrospray, triple-quadrupole), while triglycerides and cholesterol are measured by standard enzymatic colourimetric assays on a COBAS 8000 C702 analyser. In a discovery cohort (*n* = 105) and an independent validation cohort (*n* = 183), PCPro positivity (>−1.1903) correlated with a median OS reduction from approximately 24 to 12–13 months (HR 3.75 and 2.13, respectively), independent of conventional prognostic variables [87]. Building on this finding, an open-label, single-arm, multicenter phase II trial (ACTRN12622001003763) is now evaluating whether the PCSK9 inhibitor evolocumab (420 mg SC every 4 weeks for 12 weeks), when added to standard anticancer therapies (including taxane, AR-targeted therapy, olaparib, or ^177Lu-PSMA-617), can transition PCPro-positive patients to negative status. The primary endpoint is reclassification of the lipidomic signature at week 12, with exploratory assessments including safety, PSA_50_ response, and broader lipidomic shifts [88]. This trial epitomizes a strategic shift: therapeutically targeting metabolic vulnerabilities, specifically ceramide-driven resistance, in mCRPC using a precision metabolomics framework.

#### 3.2.4. Stratification Based on Immunogenic Scores and TME Signatures

The phase-2 NEPTUNES study (NCT03061539) is a non-randomised, two-cohort, biomarker-selected clinical trial, conducted at 17 UK centers, that prospectively tested whether restricting immune-checkpoint blockade to men with mCRPC exhibiting an immunogenic signature (ImS^+^) could convert an otherwise immune cold disease into one that is responsive to immunotherapy [89]. ImS^+^ was defined by at least one of: (i) mismatch-repair deficiency (MMR-d, by IHC); (ii) non-MMR DDR loss (DDR-d, by targeted exome sequencing); or (iii) high tumor-infiltrating lymphocytes (≥20% of nucleated cells, by multiplex IHC). Two sequential dosing schedules of nivolumab (NIVO) plus ipilimumab (IPI) were explored: Cohort 1 received NIVO (1 mg kg^−1^) and IPI (3 mg kg^−1^) every 3 weeks followed by NIVO (480 mg) every 4 weeks for ≤1 year, whereas Cohort 2 inverted the doses (NIVO 3 mg kg^−1^ and IPI 1 mg kg^−1^). The primary end point of week-9 composite response rate (CCR) had a prespecified success threshold of ≥40% (null ≤ 20%, one-sided α = 0.05). CRR reached 40% (14/35; 90% CI 26–55%; *p* = 0.005) with the IPI-intense schedule and 25% (9/36; 90% CI 14–40%; *p* = 0.28) with the NIVO-intense schedule, yielding an overall CRR of 32% (90%CI; 23–43). Median duration of response was 10.4 vs. 6.4 months, and median OS of 16.2 vs. 15.2 months; no OS *p*-value was reported. Grade 3–4 treatment-related adverse events occurred in 63% vs. 31% of patients (diarrhoea 42% in cohort 1). Responders were enriched for MMR-d, *BRCA1/2* loss, high TIL density, *CDK12* loss, *ATM* or *CHD1* alterations, underscoring the predictive value of a broad tumor-based immunogenic signature. Accrual and treatment are complete, and the study is in long-term follow-up [89].

While NEPTUNES leverages tumor-centric DNA-repair and T-cell density metrics to select immune hot” tumors for dual-checkpoint blockade, the next study applies a blood-based transcriptomic signature of adenosine-rich micro-environments (AdenoSig) to guide inhibition of a complementary, metabolic immune checkpoint.

In the first-in-human phase Ia/b trial (NCT02740985), 108 men with mCRPC were enrolled, of whom 65 received an Adenosine A2A Receptor Antagonist (AZD4635 monotherapy) and 43 received AZD4635 and durvalumab. Within these, 39 and 37 patients, respectively, had RECIST-measurable disease and therefore constituted the response-evaluable population [90]. The capsule formulation’s recommended phase-II dose was 75 mg once daily, whether given as monotherapy or with durvalumab (1500 mg i.v. every 4 weeks). Treatment-related grade ≥ 3 adverse events occurred in 12.9% of patients on monotherapy and 21.8% with the doublet; nausea, fatigue, vomiting, anorexia, dizziness and diarrhoea were the commonest any-grade events [90]. RECIST responses were rare with AZD4635 alone (5%, 2/39) but more frequent when durvalumab was added (16%, 6/37). Despite this modest radiographic activity, 22% of evaluable mCRPC patients (10/45) achieved a ≥50% PSA decline (PSA_50_) on the combination [90]. In conjunction with this, an adenosine-specific genomics signature, AdenoSig biomarker was employed as a baseline test for peripheral-blood profiling including 14-gene (*PPARG*, *CYBB*, *COL3A1*, *FOXP3*, *LAG3*, *APP*, *CD81*, *GPI*, *PTGS2*, *CASP1*, *FOS*, *MAPK1*, *MAPK3*, *CREB1*) [90]. Patients with “AdenoSig-high” tumors enjoyed a markedly longer median radiographic PFS (21 weeks vs. 8.7 weeks; HR = 0.46) and accounted for every PSA_50_ response recorded in the study [90]. Pharmacodynamic biopsies demonstrated on-treatment up-regulation of cytotoxic/IFN-γ transcripts (e.g., *GZMB*, *IFNG*) and down-regulation of T-regulatory gene signatures, confirming target engagement and partial reversal of adenosine-mediated immune suppression [90]. Blocking the adenosinergic checkpoint with AZD4635 is safe and shows preliminary activity in mCRPC when combined with durvalumab. Crucially, benefit is prominent in tumors classified as AdenoSig-high, illustrating how a blood-based, multi-gene signature can prospectively enrich for responders even in early-phase studies and thereby guide the development of next-generation immunometabolic combinations.

#### 3.2.5. Transcriptomic Stratification in the CHAARTED Trial

In a pre-specified correlative study nested within the phase-III CHAARTED trial (ADT and six cycles of docetaxel for metastatic hormone-sensitive prostate cancer, mHSPC), Hamid et al. profiled 160 primary-tumor blocks that passed RNA quality control (198 blocks were retrieved from the 790-patient parent trial). Tumors were categorised by the PAM50 luminal–basal classifier (luminal B 48%, basal 50%, luminal A 2%), the 22-gene Decipher genomic-classifier (GC) risk, and a 9-gene androgen-receptor-activity (AR-A) score; no neuro-endocrine signature was analysed [91]. On ADT alone, luminal B disease carried the worst prognosis (median OS of 29.8 mo vs. 47.1 mo for basal; HR = 1.75, *p* = 0.052) while basal disease responded better. Adding upfront docetaxel improved OS in the overall analysis cohort (median 53.9 vs. 32.4 mo; HR = 0.58, 95% CI 0.38–0.87; *p* = 0.009) and selectively benefited luminal B tumors (HR = 0.45, *p* = 0.007; median 52.1 vs. 29.8 months); basal tumors derived no survival advantage (HR = 0.85, *p* = 0.60). In multivariable models, high Decipher GC score (HR = 1.21 per 0.1-unit, *p* < 0.001), low AR-A, high tumor volume, and ECOG 1–2 independently predicted worse OS, whereas PAM50 subtype was not independently significant (luminal B vs. basal HR = 1.25, *p* = 0.30) [91]. Patients with GC-high (Q4) tumors experienced the largest absolute 3-year OS gain from docetaxel (25% vs. 9% in GC-low Q1 [91]. These results show that transcriptomic classifiers, particularly luminal B status and Decipher risk, refine prognosis and help identify men most likely to benefit from early docetaxel, supporting prospective biomarker-stratified treatment strategies in mHSPC [91].

#### 3.2.6. Adaptive Omics-Guided Therapy Based on Multiple Alterations

The likely most prominent example of a true prospective omics-driven stratification trial for mCRPC is the ProBio platform trial (NCT03903835) [92]. ProBio is an outcome-adaptive, Bayesian randomized study using real-time data to optimize treatment allocation probabilities based on ctDNA analysis including four known groups of omics signatures, such as (i) mutations and structural rearrangements in AR, (ii) mutations, homozygous deletions, and structural rearrangements in *TP53*, (iii) DNA-repair deficiency by detection of mutations, homozygous deletions, and structural rearrangements in *ATR*, *ATM*, *BARD1*, *CDK12*, *BRCA1*, *BRCA2*, *BRIP1*, *CHEK2*, *FANCA*, *MRE1*, *NBN*, *PALB2*, *RAD50*, *RAD51*, *RAD51B*, *RAD51C*, *RAD51D* and (iv) *TMPRSS2-ERG* fusions by structural rearrangements and deleterious events [92]. Based on the four pre-specified molecular signatures, e.g., *AR*-negative/*TP53*-wild-type, *TP53*-altered, *TMPRSS2–ERG* fusion-positive, *HRD*, along with an unselected group, patients were subsequently randomised to receive: an androgen-receptor pathway inhibitor (ARPI; abiraterone or enzalutamide), a taxane (docetaxel or cabazitaxel), or physician’s-choice control, with re-randomisation permitted at progression. In the first 218 randomisations (193 patients, some re-randomised at progression, with median follow-up of 20 months), ARPIs lengthened the composite time-to-no-longer-clinically-benefiting to 11.1 months versus 6.9 months for taxanes (survival-time ratio; STR 1.60) and improved OS of 38.7 months versus 21.7 months; the greatest ARPI advantage was observed in the *AR*-negative/*TP53-WT* and *TMPRSS2–ERG* subsets, whereas *TP53*-altered disease derived no differential benefit [92]. ProBio thus demonstrates the feasibility of real-time ctDNA profiling to drive both initial randomisation and adaptive treatment sequencing, providing a scalable template for biomarker-guided drug development in advanced PCa (Figure 4). The summary of these clinical trials is given in Table 3.

### 3.3. Miscellaneous FDA-Approved Agents with Exploratory Omics Markers

While approved therapies for advanced PCa already include specific genomic or proteomic tests for PARP inhibition and PSMA-directed radioligand therapy, a second wave of therapies based on omics stratification is rapidly approaching regulatory maturity. The five studies below exemplify how liquid biopsy dynamics, rare driver mutations, transcriptomic outliers, and peripheral immune-function assays can refine the use of agents that are already FDA-approved, placing these biomarkers one validation step away from incorporation into future prescribing information.

A prospective two-cohort study of 81 men receiving first-line enzalutamide or abiraterone (both FDA-approved) demonstrated that a simple plasma draw after 4 weeks can dichotomise patients by circulating tumor DNA (ctDNA) fraction change. Persistently detectable ctDNA conferred a four to fivefold higher risk of progression or death, whereas conversion to undetectable status aligned with durable benefit. These data support ctDNA kinetics as a real-time companion marker to trigger early intensification or switch therapy [93]. Additionally, a retrospective analysis of 634 de novo mCSPC in the Flatiron-Foundation clinico-genomic database revealed that somatic SPOP mutations (approximately 11%) were strongly predictive of benefit from androgen-receptor-pathway inhibitors (enzalutamide, abiraterone, apalutamide) but not from upfront docetaxel. Mutant cases experienced non-reached median time-to-CRPC versus 16.7 months for wild-type when treated with ARPI (HR = 0.20; 95% CI 0.06–0.63; *p* = 0.006), and a parallel OS advantage, positioning SPOP as a treatment-selection biomarker nearing prospective trial confirmation [94].

In another study investigating ERBB3 (HER3) over-expression in relation to ARPI resistance in castration-sensitive disease, bulk RNA-seq of approximately 900 treatment-naïve tumors uncovered ERBB3 over-expression in approximately 9% of castration-sensitive prostate cancers, enriched in African and Asian ancestry groups. Functional studies showed that HER3 blockade re-sensitised lines and xenografts to enzalutamide, nominating HER3 as a dual biomarker and drug target. With multiple HER3-directed antibody–drug conjugates already FDA-approved in breast cancer, a clear translational path exists for label expansion into PCa contingent on companion ERBB3 testing [95].

Finally, a two-cohort ex vivo immune-monitoring study of 134 men with mCRPC treated with the FDA-approved autologous vaccine sipuleucel-T showed that pretreatment peripheral-blood IFN-β release after TLR1/2 stimulation robustly stratifies overall survival [96]. In the 106-patient PRIME discovery set, ≥median IFN-β induction translated into an 88% reduction in mortality risk (HR = 0.12; log-rank *p* = 0.019), a benefit reproduced in an independent 28-patient KCI validation cohort (HR < 0.1; *p* = 0.047). Multivariable adjustment for age, PSA, ECOG status, and vaccine potency retained statistical significance (*p* = 0.048), positioning this functional immune-omics read-out just one validation step away from incorporation into future sipuleucel-T prescribing information [96].

### 3.4. Prospective Trials for Predictive and Prognostic Markers in Advanced PCa

In a prospective, single-arm, multicentre Phase II “CabaBone” study conducted by the Hellenic Cooperative Oncology Group 60 men with bone-dominant mCRPC were enrolled to receive cabazitaxel 25 mg m^−2^ every three weeks plus daily prednisone. At the pre-specified landmark, the 6-month progression-free-survival (PFS) rate was 47% (95% CI 33–59) and the 12-month OS rate was 70% (95% CI 56–80) [97]. Because the protocol did not include formal one-sample hypothesis testing, *p*-values for these primary efficacy end-points were not reported. Exploratory correlative analyses demonstrated that the presence of reactive haematopoiesis on baseline bone-marrow biopsy was associated with significantly longer PFS (log-rank *p* = 0.04) and OS (*p* = 0.02), whereas HRR gene mutations, detected in 12% of tumors, were not prognostic (*p* > 0.10). Treatment-associated toxicity was consistent with the established cabazitaxel safety profile, and no new grade ≥ 3 adverse-event signals were observed. Collectively, these data reinforce cabazitaxel’s clinical activity in bone-metastatic mCRPC and illustrate the feasibility of embedding systematic tumor, bone-marrow, and liquid-biopsy collection within pragmatic Phase II designs to refine patient selection [97].

Feng et al. undertook a pre-specified transcriptomic correlative study within the randomised, double-blind phase-III SPARTAN trial to determine whether genomic classifiers modulate the efficacy of apalutamide in non-metastatic castration-resistant prostate cancer (nmCRPC) [98]. Archived diagnostic tumor samples from 233 of the 1207 participants (in a two-to-one ratio of apalutamide vs. placebo) passed micro-array quality control. Whole-transcriptome exon arrays were used to assign each case a Decipher genomic-classifier (GC) score (high-risk > 0.6 vs. low/average ≤ 0.6) and a PAM50-derived basal-luminal subtype. Half of the profiled tumors were high-risk GC and 65% were basal [98].

Stratified analyses showed the largest incremental benefit in the high-risk GC group (MFS HR = 0.21, 95% CI 0.11–0.40; *p* < 0.001), which translated into improvements in OS (HR = 0.52, 95% CI 0.29–0.94; *p* = 0.03) and second progression-free survival (PFS_2_ HR = 0.39, 95% CI 0.23–0.67; *p* = 0.001). Lower-risk GC tumors also benefited, but less markedly (MFS HR 0.46, 95% CI 0.23–0.95; *p* = 0.04) [98]. Apalutamide prolonged MFS in both basal (HR 0.34, 95% CI 0.20–0.58; *p* < 0.001) and luminal tumors (HR = 0.22, 95% CI 0.08–0.56; *p* = 0.002); within the apalutamide arm, luminal subtype conferred a further advantage over basal disease (HR = 0.40, 95% CI 0.18–0.91; *p* = 0.03). Multivariable Cox modelling confirmed that both GC risk category and basal-luminal status independently predicted the outcome on apalutamide, indicating that patients with high-risk GC or luminal tumors derive the most durable disease control and survival benefit from early androgen-receptor-pathway intensification in nmCRPC.

Along these lines, in a prospective, multicentre phase II study (*n* = 65) integrating mandatory baseline and progression metastasis biopsies plus serial CTC (Circulating Tumor Cells) profiling, enzalutamide resistance in mCRPC proved highly heterogeneous [99]. Whole-exome sequencing showed that, compared with baseline, progression biopsies were enriched for AR amplifications (64.7% vs. 53.9%; *p* < 0.05) and newly acquired *BRCA2* alterations (64.7% vs. 38.5%), while losses in *PTEN*, *RB1*, and *TP53* were frequent at both time-points. CTC RNA-seq at progression revealed up-regulation of AR splice variants, AR-regulated transcripts, and neuro-endocrine markers; this “resistant” CTC signature was associated with markedly shorter OS (HR = 6.3; *p* = 0.01). The data underscore the dynamic evolution of AR-pathway and DNA-repair lesions under enzalutamide and support serial liquid-plus-tissue sampling to guide post-ARPI therapeutic choices [99].

Routine genomic profiling followed by multidisciplinary molecular tumor board review was evaluated for 277 men with advanced PCa treated prospectively in 2017–2020 at a tertiary Dutch center [100]. Metastatic biopsies underwent hybrid-capture NGS; 215 cases (78%) reached the molecular tumor board review. Actionable variants appeared in 102 patients (47%), generating a genetically matched-therapy (GMT) advice, PARP inhibitors for *BRCA1/2* or *ATM* loss (62%), PD-(L)1 blockade for MSI-high/TMB-high (18%), or kinase/PI3K agents (16%). Sixty-three patients (62% of those advised) initiated GMT after a median of 42 days; the main obstacles were rapid clinical decline and trial ineligibility. In the treated group, RECIST responses occurred in 38.5%, ≥50% PSA declines in 43.5%, and median progression-free survival was 5.3 months, with 41% progression-free at 6 months. These real-world data show that systematic sequencing can deliver clinically meaningful precision therapy for PCa outside trials [100].

An ancillary biomarker study of the STAMPEDE abiraterone comparison profiled 781 men at baseline before starting androgen-deprivation therapy (ADT) and/or abiraterone/prednisolone [101]. Using a CLIA-compatible whole-transcriptome assay, the investigators computed 57 published expression signatures, including Decipher (commercial 22-gene prostate-cancer genomic classifier), PAM50–luminal, proliferation, *PTEN*-loss, and *TP53*-loss scores, and tested these against failure-free and OS in multivariable Cox models. Decipher high-risk (>0.60) was the strongest adverse prognostic marker and identified non-metastatic (M0) patients who gained the largest absolute survival benefit from abiraterone intensification. In metastatic disease, high proliferation, *PTEN*-loss, *TP53*-loss, and “treatment-persistent cell” signatures remained independently detrimental, whereas androgen-receptor activity was protective only in M0 tumors. A dense interferon-β inflammatory signature correlated with lymphocytic infiltration yet predicted worse outcome. Although none of the signatures were universally predictive, these prospective data show that multi-gene RNA can stratify ADT responders and help prioritise early abiraterone, warranting external validation and formal regulatory qualification [101].

Additionally, IMbassador 250 was a global, open-label, randomized phase III trial that tested whether adding the anti-PD-L1 antibody atezolizumab to enzalutamide could improve survival in mCRPC patients who had progressed after abiraterone/prednisone [102]. 759 men were allocated in a one-to-one setting to receive atezolizumab (1200 mg q3 wks) and enzalutamide (160 mg qd) versus enzalutamide alone. The study did not meet its primary end-point: median OS was 15.2 months in the combination arm versus 16.6 months with enzalutamide alone (HR = 1.12; 95% CI 0.91–1.37; *p* = 0.28). rPFS survival and PSA responses were likewise unchanged, and Grade ≥ 3 all-cause adverse events occurred in 54% vs. 35% of patients (treatment-related grade 3/4: 28% vs. 10%) [102].

Although no prospective biomarker stratification was built into the design, extensive correlative work was performed to assess: (i) PD-L1 expression on immune cells (IC2/3 by SP142 IHC), (ii) a tumor T-effector interferon-γ gene signature using the HTG EdgeSeq (Hybridization-based Targeted Gene Expression Sequencing) assay on archival FFPE tissue, and (iii) high intratumoral CD8^+^ T-cell density [102]. Each was associated with numerically longer OS in the atezolizumab arm, whereas patients lacking these features derived no benefit. Tumors harbouring DDR mutations (*BRCA2*, *ATM*) also showed a non-significant trend toward improved outcome with the combination, but small numbers precluded firm conclusions. From an omics-guided-trials perspective, the molecular assays were purely exploratory, were applied post hoc, and did not direct treatment allocation. Nevertheless, the above biomarker work underscores once more the need for adopting prospective transcriptomic or genomic selection (for example, the ImS^+^-guided NEPTUNES study), rather than enrolling unselected mCRPC populations [102].

Recent liquid-biopsy analysis confirms that the quantity and complexity of circulating-tumor DNA independently stratify risk in mCRPC. Huang et al. applied low-pass whole-genome sequencing to baseline plasma from two unselected mCRPC cohorts (*n* = 180) [103]. An 11-gene copy-number, alteration (CNA) score incorporating *AR*, *MYC*, and *TP53* divided patients into high- and low-risk groups. High CNA burden translated into a ≥2-fold shorter OS in both the PROMOTE abiraterone trial (*p* = 0.00019) and an independent Mayo registry (*p* < 0.0001) and predicted inferior radiographic PFS on abiraterone. Early declines in *AR* CNAs tracked clinical responders, suggesting on-treatment monitoring utility [103]. Shaya et al. retrospectively reviewed commercial Guardant360/Tempus xF reports for 63 men ctDNA was detectable in >90% of samples; harbouring > 1 pathogenic alteration conferred a seven-fold higher mortality risk of median OS 8.8 vs. 26.1 months (multivariable HR = 7.0, 95% CI 2.2–23.1; *p* < 0.001). Frequent alterations involved *AR*, *TP53*, and clinically actionable *HRR/MMR* defects appeared in 16% [104]. Jayaram et al. conducted a prospective omics-enabled biomarker analysis within a phase-II abiraterone study, employing targeted ctDNA sequencing (genomics) to show that early ‘gene-conversion’ loss of detectable *TP53*, *RB1*, or *PTEN* alterations between baseline and cycle-2 marked patients with improved survival, whereas persistent alterations portended resistance [105]. Because therapy itself was not adapted to the genomic read-out, the work exemplifies an omics-based prognostic study [105]. Together, these studies show that both genome-wide CNA load and simple ctDNA mutational metrics deliver robust, non-invasive prognostic information and support incorporating ctDNA burden into future biomarker-stratified trials.

Pan et al. prospectively profiled 141 men with mCPRC using three orthogonal assays, NGS of plasma ctDNA for HRR defects, dual-tracer ^68Ga-PSMA/^18F-FDG PET CT for a PSMA-high/FDG-negative imaging signature, and *PTEN* immunohistochemistry, and completed all tests in 106 patients [106]. Actionable findings were common: 34/106 (32%) were PSMA-high/FDG-negative, 30/106 (28%) harbored pathogenic HRR alterations (principally *BRCA2*, *ATM*, *CDK12*), and 16/106 (15%) showed *PTEN* loss; overall, 64/106 (60%) carried at least one targetable abnormality. In the subset of 74 men starting first-line abiraterone, either a PSMA-high/FDG-negative scan or an HRR defect independently predicted earlier progression (median PFS 6.8 vs. 11.2 months, *p* = 0.011, and 6.6 vs. 12.3 months, *p* = 0.002, respectively), whereas *PTEN* status had no effect. Multivariable analysis confirmed that the presence of any actionable marker conferred a two-fold higher risk of progression (adjusted HR = 1.95, 95% CI 1.14–3.34). These data show that the majority of mCRPC patients are eligible for precision-medicine trials yet gain limited benefit from unselected AR-targeted therapy, underscoring the need for prospective, biomarker-directed treatment allocation in forthcoming clinical studies [106].

### 3.5. Omics Driven Drug Development: Drug Repurposing Paradigm

In this section, we review studies that systematically identify drug-repurposing strategies and novel therapeutic targets via integrative laboratory and computational approaches rather than clinical trials or approved therapies alone. We categorize the evidence into three methodological groups: (i) genomics/pharmacogenomics, including somatic and germline variants predicting treatment efficacy and toxicity; (ii) transcriptomic signatures and network-based repurposing, including bulk/single-cell RNA signatures and interactome-based analyses linking expression profiles to candidate drugs; and (iii) multi-omics, AI, and network-level drug prediction, including platforms integrating epigenomic, proteomic, CRISPR-dependency data, and machine learning to prioritize drugs and biomarkers. Each study is summarized with emphasis on its discovery pipeline, identified targets, repurposed agents, and limitations, facilitating a clear understanding of how multi-dimensional omics data translate into clinically actionable hypotheses.

#### 3.5.1. Genomics/Pharmacogenomics-Based Repurposing

Paired whole-genome (*n* = 45) and transcriptome (*n* = 31) sequencing of metastatic biopsies collected before and after first-line androgen receptor signalling inhibitor (ARSI) therapy allowed Zhu et al. to map evolutionary routes to resistance [107]. Copy-number profiling revealed a convergent gain spanning a 14-kb enhancer downstream of *AR*, while somatic single-nucleotide-variant analysis identified recurrent ligand-binding-domain substitutions (L702H, H875Y, T878A). Integrative clustering of the RNA-seq data produced a resistance signature marked by *SSTR1* down-regulation and a shift toward AR-indifferent lineage states. Tumors with low *SSTR1* expression in an external SU2C cohort derived significantly less benefit from subsequent ARSIs, and the approved somatostatin-receptor agonist pasireotide reversed the proliferative phenotype in *SSTR1-low* 22Rv1 cells, nominating it for drug repurposing [107].

In a functional-genomics target-discovery study, Raut et al. overlaid differential- expression profiles from 208 SU2C castration-resistant prostate cancers and 52 normal prostates with DepMap CRISPR-loss fitness scores, The Cancer Genome Atlas validation, Gleason grade and overall-survival data [108]. Of the resulting 48-gene kinase/phosphatase module, *MARK3* uniquely satisfied all filters over-expression (log_2_FC > 1, false discovery rate < 0.01), adverse prognosis and CRISPR dependency. Pharmacological inhibition with PCC0208017 suppressed a *MARK3* response signature enriched for androgen-response, epithelial–mesenchymal transition, mTOR and MYC pathways, inducing G1 arrest and blocking migration in 22Rv1 cells [108].

Finally, the prospective multicenter ABIGENE trial genotyped 13 candidate single nucleotide polymorphisms in *CYP17A1*, *SLCO2B1* and *SLCO1B3* among 137 abiraterone-treated metastatic patients (median follow-up 56 months) [109]. The *CYP17A1* rs2486758 C/C genotype independently predicted shorter 3-year biological progression-free survival (HR = 4.05; 95% CI 1.46–11.22; *p* = 0.007), whereas *CYP17A1* rs743572 C/C and poor ECOG performance status independently forecast grade ≥ 3 toxicity. These findings illustrate how constitutional pharmacogenomics can complement tumor-omics in guiding therapy selection [109].

Collectively, these studies showcase how tumor genomics, functional-omics triage, and germline variation converge to reveal drug-repurposing avenues and toxicity predictors while underscoring the need for larger cohorts and multi-model validation before clinical translation.

#### 3.5.2. Transcriptomic Signature & Network-Based Repurposing

Publicly available bulk-RNA-seq datasets of CRPC were mined by Golla et al. to derive a gene signature enriched for extracellular-matrix components and cytoskeletal regulators, including *COL3A1*, *FN1*, *ACTN1*, *MYH4*, and *CALR* [110]. Protein–protein interaction mapping pinpointed two high-degree hubs, the efflux transporter ABCC4 and the prostate-specific membrane antigen FOLH1, as tractable therapeutic entry points. Structure-based virtual screening of DrugBank-approved compounds ranked flutamide as a top ligand for the ABCC4 nucleotide-binding pocket and N-acetyl-*D*-glucosamine as a favourable binder of the FOLH1 catalytic site [110].

Chang et al. interrogated two bone-metastatic CRPC micro-array series (GSE32269, GSE77930), identifying a 229-gene extracellular-matrix/integrin program and, using ESTIMATE, ssGSEA/EPIC, and CIBERSORTx, confirmed a micro-environment dominated by M2 macrophages [111]. Weighted-gene-co-expression analysis coupled with random-forest classification focused on a 16-gene core centered on *SPP1* and *COL11A1*. Projection onto the interactome and triple Connectivity-Map screening produced 62 candidate chemotypes; OncoPredict prioritised docetaxel and navitoclax, whereas ADMET filters highlighted foretinib, norethindrone, testosterone, and menthol, alongside docetaxel, for immediate testing. Molecular docking followed by 10-ns molecular-dynamics simulations suggested that mulberroside C and terrestrosin D may disrupt SPP1–integrin/CD44 interfaces. While comprehensive, the pipeline remains computational and awaits validation in bone-tropic CRPC models [111].

In another transcriptomics-based study where scIDUC, an algorithm that embeds single-cell RNA-seq profiles into a pharmacogenomic latent space (CTRPv2, GDSC-2), accurately stratified drug-resistant versus sensitive sub-clones across rhabdomyosarcoma, pancreatic ductal adenocarcinoma, and CRPC [112]. In CRPC, it predicted and viability assays on docetaxel-resistant DU145 cells confirmed (*n* = 3; two-way ANOVA *p* < 0.0001) a selective vulnerability to the BRAF inhibitor vemurafenib, outperforming CaDRReS-Sc and Beyondcell [112]. The study illustrates how single-cell expression signatures can be coupled to large drug-response atlases to generate clone-specific repurposing hypotheses, although broader validation across additional CRPC lines and in vivo models is still required [112].

Cyclosporin A (CsA), 10 µM, 24 h, differentially expressed 3319 genes in PC-3 cells compared with vehicle controls (GSE109505); 2500 were down-regulated and 819 up-regulated. Of these, 871 showed an inverse correlation with the metastatic-CRPC signature and were enriched for cell-cycle pathways [113]. ARACNe and Bayesian network analysis converged on E2F8 as a master regulator downstream of the kinase MELK, both repressed by CsA. Silencing gene curtailed proliferation in four CRPC lines (*p* < 0.005), and 20 mg kg^−1^ CsA reduced 22Rv1 xenograft volume by 45% (*p* < 0.05). High E2F8 expression (HR = 3.03, *p* = 0.0002), particularly when co-elevated with *AR* (HR = 4.33, *p* = 0.0019), predicted poor survival in TCGA-PRAD, placing the long-used immunosuppressant as a repurposing candidate that disrupts the MELK/E2F8 axis [113].

In another study where authors mined TCGA data, isolating 319 prostate-cancer DEGs enriched in xenobiotic-metabolism pathways, network analysis pinpointed AMACR, FOLH1, and NPY as core hubs. CIBERSORT-based deconvolution linked FOLH1 to heightened CD8^+^-T-cell infiltration and showed all three hubs inversely associated with CD4^+^-T cells, implicating them in shaping the tumor immune milieu [114]. A second comparison (hormone-sensitive vs. castration-resistant PCa) yielded 426 DEGs; lipid-metabolic circuits (arachidonic-acid, PPAR, AMPK) dominated, with SCD and FASN emerging as progression-specific targets. Connectivity-Map screening nominated aminoglutethimide and resveratrol as candidate repositioned drugs, collectively positioning metabolic enzymes (AMACR, SCD, FASN) and FOLH1 as actionable biomarkers for precision therapy. Two studies tackled cabazitaxel resistance via signature-reversal screens [114].

Hongo et al. established cabazitaxel-resistant PC3CR cells from parental PC-3 by chronic drug exposure and used whole-transcriptome profiling to define a resistance signature, which they queried against the Broad Connectivity Map to nominate etoposide a Topoisomerase II α inhibitor, the top candidate to reverse the resistant program [115]. Independently, a 257-gene spindle-regulator signature from DU145-CR/PC-3-CR cells nominated the antipsychotic pimozide. Pimozide down-regulated *AURKB* and *KIF20A*, synergised with cabazitaxel, and shrank DU145-CR tumors, linking mitotic-checkpoint attenuation to drug resensitisation [116].

Multi-omic profiling of cabazitaxel-refractory tumors uncovered oxytocin-receptor (*OXTR*) signalling as a tumor-cell niche. Single-cell RNA-seq of circulating tumor cells and 10× Visium spatial transcriptomics identified *OXT/OXTR* pathway activation; an in silico FDA-drug screen flagged the antitussive cloperastine, which inhibited DU145-CBZ-resistant growth, synergised with cabazitaxel (combination index < 1), and reduced xenograft burden while suppressing pathway-phosphorylation sites [117].

Across five bulk-RNA-seq cohorts (*n* = 905), Huang et al. built a 47-gene replication-stress signature (RSS) that out-performed Gleason score and PSA (TCGA 5-yr AUC = 0.864). RSS-high tumors harbored *TP53/RB1/PTEN* loss and immune exclusion. Linking RSS to GDSC drug matrices prioritised TOP1 inhibitors (irinotecan, topotecan); siRNA or pharmacological blockade of RSS drivers *FEN1* and *RFC5* induced apoptosis in C4-2B and PC-3 cells, experimentally validating the model [118].

Glucocorticoid-treated, ARSI-resistant epithelial cells and cancer-associated fibroblasts shared a four-gene glucocorticoid-receptor (GR) core in which monoamine-oxidase-A (MAO-A) was the dominant direct GR target [119]. *MAO-A* up-regulation paralleled stronger GR/AR activity in explants and patient biopsies and predicted earlier progression. Two licensed MAO-A inhibitors, clorgyline and phenelzine, reduced proliferation and synergised with enzalutamide, abiraterone, and docetaxel, offering an immediately testable strategy to blunt GR-mediated escape [119].

Integration of bulk (TCGA-PRAD, GSE70769) and single-cell (GSE245387) transcriptomes revealed a high-pyrimidine “P2” subtype driven by nucleotide-biosynthesis genes; machine-learning convergence nominated *RRM2*, whose over-expression worsened disease-free survival (*p* = 0.008) [120]. Drug-response modelling predicted greater sensitivity of P2 cells to BI-2536, PAC-1 and related agents, emphasising nucleotide metabolism as vulnerability.

The GSFM pipeline converts noisy drug-perturbation profiles into functional-module scores; its composite RS_GSFM_ correlated with IC_50_ across three cell lines (ρ = 0.50–0.72, *p* < 0.001) and identified NTNCB as a nanomolar inhibitor of PC-3 viability that reduced xenograft mass by 67% without weight loss [121].

Finally, a meta-analysis of six GEO series plus TCGA-PRAD, SU2C/PCF, and MSK-IMPACT RNA-seq yielded two consensus signatures: PCa-sig (primary vs. metastatic, AUC = 0.93) and NE-sig (AR-positive vs. neuro-endocrine CRPC, AUC = 0.91). Overlay with DepMap/Project-SCORE essentials nominated *EZH2*, *AURKB*, *RRM2*, *HSPD1*, *BOP1*, and *MCM10*; drug-sensitivity mapping prioritized Panobinostat, vorinostat, dabrafenib, trametinib, and 17-AAG for context-selective evaluation [122].

Together, these studies broaden the transcriptomic repurposing landscape from risk-group stratification and resistance reversal to pathway-centric vulnerabilities, while underscoring the need for multi-model validation and early-phase trials to convert in silico leads into a therapeutic reality.

#### 3.5.3. Multi-Omics/AI and Network-Level Drug Prediction

Recent efforts extend beyond single-omic transcriptome mining to truly integrative platforms. An integrative, multi-omics framework from He et al. combined bulk transcriptomics (838 tumors) and single-cell datasets from 12,401 epithelial cells derived from seven patients (three primary and four with CRPC) to generate a 14-gene machine-learning-based Histone-Modification Score (termed CMLHMS) [123]. High CMLHMS conferred a three-to-four-fold higher risk of biochemical recurrence (pooled HR = 3.72, 95% CI 2.67–5.20; *p* < 0.001) and, via CMap, predicted preferential sensitivity to PI3K and EGFR inhibitors, whereas low-score tumors favoured taxanes or gemcitabine [123].

Complementing this, Bacolod et al. re-analysed 209 normal–primary–metastatic samples together with DepMap CRISPR screens and PRISM drug viability data. DNA-replication/PLK1 programmes dominated metastatic transition, and the FDA-approved multi-kinase inhibitor fostamatinib emerged as preferentially active in invasion-high cell lines. The pipeline simultaneously nominated surface (ADAM15, CD276) and secreted (APLN, ANGPT2) markers as minimally invasive diagnostics [124].

Rydzewski et al. introduced TARGETS, an elastic-net model trained on GDSC multi-omics that retained significance for all 18 overlapping drugs in CCLE (FDR < 0.05). Across 9430 TCGA tumors, TARGETS faithfully recapitulated every FDA-approved biomarker–drug pairing (e.g., *EGFR*-mutant LUAD with EGFR TKIs, *p* < 0.0001; MGMT-methylated GBM with temozolomide, *p* < 0.0001) [125]. Clinical validation in 100 WCDT mCRPC biopsies confirmed that the AR-signalling-inhibitor score predicts 51–100% PSA responses to enzalutamide/abiraterone, with a significant treatment-score interaction (*p* = 0.0252). Collectively, these studies exemplify how multi-layer data fusion and machine learning can surface both drug targets and readily repurposable agents for advanced PCa [125].

OncoLoop pairs patient RNA-seq profiles with “cognate” genetically engineered mouse–derived tumors (GEMM-DTs), then mines drug-perturbation libraries for compounds that invert shared Master-Regulator activity. In prostate models, OncoLoop-nominated drugs were validated in allograft, syngeneic, and PDX systems and potentiated standard agents, including the PD-1 blocker nivolumab and the AR inhibitor enzalutamide, highlighting a scalable route from transcriptome to combination therapy [126].

These four exemplars show how converging data layers, machine learning, and regulatory-network analytics can produce clinically testable, subtype-specific drug hypotheses, yet each still awaits prospective, biomarker-driven trials to confirm real-world benefit in advanced prostate cancer.

## 4. Discussion

This review compiles evidence demonstrating that molecular stratification is no longer an experimental addition in advanced PCa but rather a prerequisite for optimal therapy selection. PARP inhibitors, either as monotherapy or in combinatorial therapies, demonstrate benefit within *BRCA1/2*- and/or HRR mutation-bearing tumors [56,59,60]. Moreover, PSMA-directed radioligand therapy with ^177Lu-PSMA-617(VISION) is another example of FDA-approved biomarker-guided therapies. Additionally, PARP blockade is the second FDA-approved, companion diagnostic-mandated option to independently predict survival [71]. A summary of the most promising biomarkers associated with therapeutic pharmacological intervention, either already approved or in running clinical trials, is given in Table 4.

At the same time, multiple reports shed light on omics-based biomarkers that are applied as stratification means within running clinical trials (summarized in the second section of this article and Table 3). An exemplary case of an adaptive clinical trial where omics biomarkers are guiding treatment is the ProBio trial, where ctDNA biomarker-based screening is used to randomize patients with advanced PCa into four treatment arms, with the possibility to adapt therapy based on the biomarker status [92].

Despite the evident progress in the field, as summarized also in Table 4, there are still several challenges that hinder the wide applicability of omics-mediated therapeutic schemes: (i) on the one hand related to the heterogenic nature of advanced PCa, (ii) secondly, associated with technical limitations of the biomarker testing and (iii) lastly, with inherent limitations of targeted (and even more combination thereof) therapies.

Firstly, mCRPC presents with intratumor heterogeneity, whereby molecularly distinct subclones coexist within the same tumor, or across metastatic sites. This spatial and temporal heterogeneity contributes to significant variability, particularly in the mutational status when comparing primary to metastatic sites [127]. This becomes a major clinical challenge when treating patients with advanced metastatic disease, based on a mutation screening that was performed in primary and/or archival tissue blocks. Depending on the targeted mutation, concordance varies relatively high, though not absolute, concordance exists between primary prostate tumors and metastatic sites for pathogenic DDR alterations. In the largest paired-sample study to date, 84% of DDR mutations detected in metastatic tissue or ctDNA were already present in the corresponding archival primary specimen (95% CI 71–92%), indicating that most are truncal events acquired early in tumor evolution [58]. Real-world datasets confirm this, showing moderate to substantial agreement between primary–ctDNA (κ = 0.59) and metastatic–ctDNA (κ = 0.65) pairs, and suggesting that specimens are largely interchangeable for actionable DDR variants [128]. Archival primary tissue is an appropriate first-line specimen for DDR and mismatch-repair (MMR) testing in mCPRC. However, negative, outdated, or equivocal results should trigger reflex testing of a contemporary metastatic biopsy or high-tumor-fraction ctDNA, particularly at progression or after exposure to targeted agents such as PARP inhibitors [129]. Nevertheless, therapy-induced variants, such as multiclonal *BRCA2* reversion mutations or acquired AR alterations, have been identified exclusively in post-treatment metastatic biopsies or ctDNA in up to one-third of advanced PCa cases, reflecting both spatial and temporal heterogeneity. Reliance solely on the primary tumor in such contexts, therefore, risks missing emergent resistance mechanisms and may lead to unexpected treatment resistance [130]. To address such intra-patient heterogeneity in mCRPC, emerging strategies such as serial liquid biopsies, multiregional sampling, and longitudinal genomic profiling are applied to capture clonal evolution and resistance mechanisms more effectively [127,131]. These approaches help overcome sampling bias and improve the detection of subclonal mutations, allowing for more accurate patient selection and adaptive treatment strategies [132]. However, challenges remain in standardizing these methods and integrating them into routine clinical workflows.

An additional challenge of the omics-guided treatment approaches is associated with archival tissue analysis (particularly concerning NGS). Archival tissue blocks are frequently characterized by low tumor content and degradation of the biological material. The importance of high-quality *BRCA1/2* testing for guiding treatment with PARP-inhibitor was underscored within a recent multicenter study [133]. Sequencing success in 954 prostate-cancer FFPE specimens fell from 88% in blocks < 1 year old to 69% when storage exceeded two years, with DNA concentration and fragmentation emerging as independent predictors of failure. These real-world data support policies that prioritize fresh biopsies or rapid testing to avoid missing PARP-eligible patients [133].

Liquid-based biopsy, as one of the proposed solutions to the above problem, is promising, but it is still evolving with its routine use in broader clinical practice to await further validation. Liquid biopsy-based markers are incorporated into clinical trials for advanced PCa to assess eligibility for specific therapies, monitor treatment response, assess minimal residual disease (MRD), and assess response to PARPi or androgen receptor signaling inhibitors [134]. Moreover, blood or serum-based liquid biopsies are generally favored for their ease of sampling and ability to capture disease heterogeneity [135]. They are used as a complement to traditional biopsy, especially when tissue is difficult to access or not available [135]. One important factor is that currently, concordance between tissue and liquid biopsy in detecting HRR and other actionable mutations is moderate and context-dependent, as mentioned above, largely depending on the mutation type. Such moderate concordance underscores the complementary nature of these approaches rather than their interchangeability. Due to this, liquid biopsy is often considered a useful adjunct but not a standalone replacement for tissue testing in many clinical scenarios [136].

Importantly, cost-effectiveness of targeted treatment approaches, exemplified by PARP inhibitors and PSMA-targeted radioligand therapy, presents a complex and often contentious landscape. For example, PSMA radioligand therapy meets Germany’s willingness-to-pay threshold at about €69,000 per quality-adjusted life year (QALY) but falls short in countries with stricter benchmarks based on models where only health-system costs are considered. When patient out-of-pocket expenses and productivity losses are added (“societal perspective”), even PSMA therapy often fails to demonstrate acceptable value in many high-income settings [137]. PARP inhibitors face even steeper economic hurdles, as their high cost result in incremental cost-effectiveness ratios above conventional cutoffs, approximately CAD$565,000/QALY in Canada and AU$144,000/QALY in Australia [138,139]. Within this context, treatment with a PARP inhibitor such as olaparib for a year can cost between US$100,000 and US$150,000 in the United States and around €5000 monthly (approximately €60,000 yearly) in Germany, underscoring the financial challenges associated with accessing these therapies.

Despite these challenges, the field is evolving with already successful paradigms paving the way. Certainly, omics screens offer unique possibilities because of the complex and high-dimensional nature of the data (profiles). As omics data evolve, we envision that integration of multiple omics datasets, and/or molecular signatures (networks or pathways) will improve the prediction of therapy response, stratify patients more accurately, and subsequently more accurately guide therapy. We would also expect that an improved understanding of the molecular pathophysiology of the advanced PCa (through multiple molecular layers) can fill an existing clinical need for predicting side effects, which are particularly pronounced after patients receive a combination of targeted therapies. At the same time, this requires standardization and alignment of the standard operating protocols and laboratory procedures, and characterization of the variability that can be introduced based on the measurements in different analytical platforms, along with validation of analytical performance according to regulatory requirements [140]. In parallel, harmonization of heterogeneous data formats, normalization methods, and annotation standards [141] are becoming a pre-requisite for the implementation of such multi-omics approach.

Besides the advancements at the molecular level, AI-based tools are also increasingly used: First, multimodal large language models (LLM) that can incorporate a variety of modalities, including across pathology slides, imaging, genomics, and by guidelines, now achieve up to 80% decision accuracy in simulated tumor-board cases, vastly outperforming standalone LLMs [142]. Second, early studies within collaborative European cancer research network such as EUCAIM (EUropean Federation for CAncer Images), OPTIMA (Optimal Treatment for Early-stage Prostate Cancer) and other networks suggest that federated models (machine-learning systems trained across multiple sites, pooling the raw data in one place) match or modestly exceed public, centrally trained models; recent demonstrations involve >60 institutions, but further large-scale validation is still in progress [143]. Third, AI-driven feature extraction radiomics, pathomics, and longitudinal ctDNA-omics already predict immunotherapy response with up to 89% (AUC 0.98–1.00) accuracy and are being embedded in adaptive trial designs [144]. Moving forward, complementing existing therapeutic approaches with the above multi-parametric models could lead to true precision oncology and improved treatment of advanced PCa.

## 5. Outlook

Our synthesis integrates five omics dimensions and explicitly links them to therapy development stages, thereby providing the first complete overview that spans approved drugs, investigational combinations, and computational repurposing. Previous narrative reviews have typically focused on a single omics layer, genomics-only surveys of PARP inhibition, or meta-analyses of PSMA radioligand therapy. The summarized results are based on a search confined to Web of Science and publications from 2021 onward, so earlier or non-indexed studies may have been missed. Additionally, the analyses relied on aggregate study-level data, preventing formal examination of within-study heterogeneity or predefined subgroup effects, while the risk-of-bias ratings were based solely on information reported in each study. Under these considerations, this article provides a comprehensive view on where we stand in terms of omics-guided therapies in advanced PCa, as well as a perspective on a “test-first-treat-later” paradigm. Both up-front germline and somatic DNA-repair profiling and PSMA-PET/CT imaging are included in the latest ASCO guidelines for men with mCPRC disease [145]. Serial liquid biopsies add a dynamic dimension, allowing early intensification or therapy switching rather than waiting for reaching endpoints like PFS [135]. Importantly, several studies show that prognostic assays need not be predictive: CTC enumeration and ctDNA fraction independently stratify risk even when they do not direct treatment choice, providing a clear numerical reference point for shared decision-making [135,136]. Taking this together, despite the technical and analytical challenges, the field is rapidly expanding, with multiple omics layers expected to improve our understanding of underlying molecular pathology and thus improve therapeutic guidance in disease stages that are still difficult to treat.

## Figures and Tables

**Figure 1 ijms-26-07475-f001:**
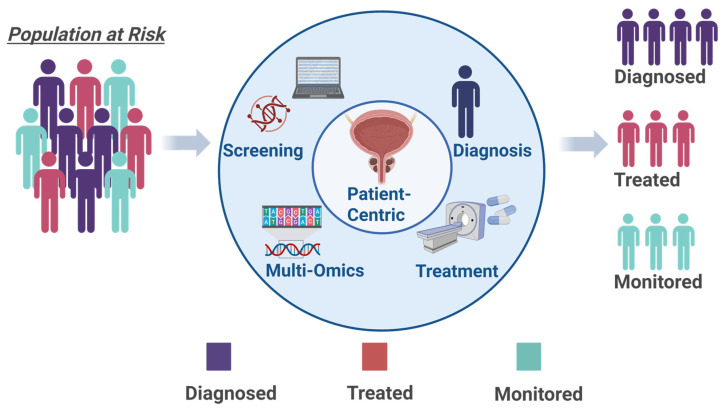
Areas for application for omics-mediated intervention: a patient-centric oncology workflow for patients with PCa. A population at risk is funnelled through four omics-informed settings, screening, diagnostic, molecular stratification, and therapy selection, while the patient remains the focus. Colour-coded icons track the care continuum: purple for diagnosed, pink for treated, and teal for monitored. Together, they illustrate how integrating genomics, proteomics, radiomics, and other -omics data personalises management from early detection to long-term surveillance. Created in BioRender. Fatima, Y. https://BioRender.com/iap7qvo (accessed on 30 June 2025).

**Figure 2 ijms-26-07475-f002:**
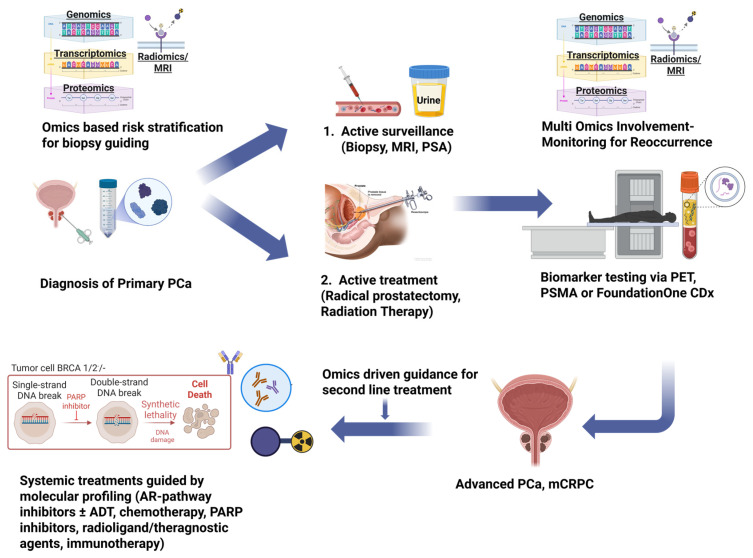
Evolution of PCa management towards precision medicine. Starting at diagnosis, next-generation profiling that combines genomics, transcriptomics, proteomics, and radiomics (top left) guides biopsy decisions and risk stratification. Depending on the molecular risk, patients enter either (1) an active-surveillance pathway (serial PSA, MRI, and repeat biopsy; top center) or (2) receive definitive local therapy with radical prostatectomy and/or radiotherapy (middle left). Post-treatment, multi-omics surveillance, including PSMA-PET imaging and comprehensive tissue or liquid-biopsy panels such as FoundationOne CDx, is used to detect molecular or radiological signs of recurrence (top right). Progression to advanced disease (bottom right) triggers a second round of omics-guided selection of systemic therapies: DNA-repair defects (e.g., *BRCA1/2* loss) nominate PARP inhibitors; PSMA expression enables lutetium-177-PSMA-617 radionuclide therapy; and other actionable alterations similarly funnel patients to mechanism-matched agents (bottom left). Created in BioRender. Fatima, Y https://BioRender.com/2550wdt (accessed on 30 June 2025).

**Figure 3 ijms-26-07475-f003:**
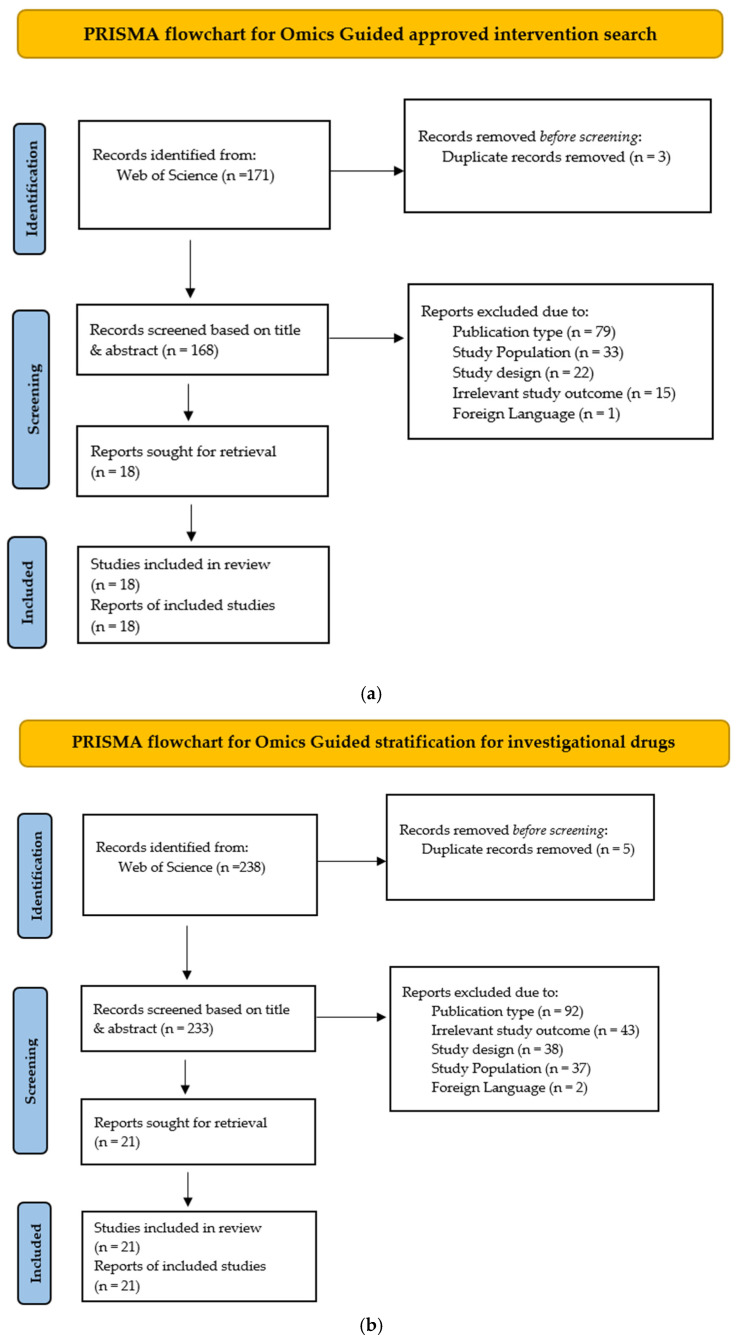
(**a**) PRISMA diagram for approved treatment search. (**b**) PRISMA diagram for treatments in interventions in Clinical trials. (**c**) PRISMA diagram for drug repurposing search.

**Figure 4 ijms-26-07475-f004:**
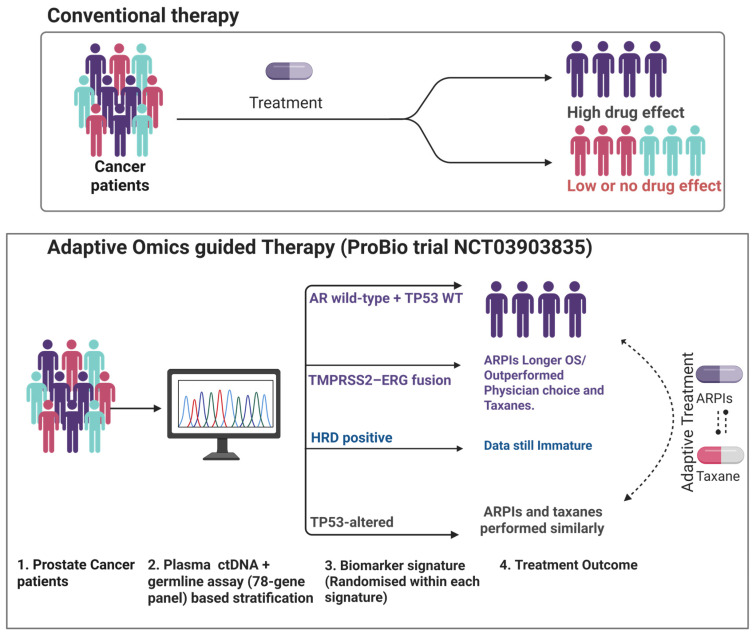
ProBio outcome-adaptive platform trial schema (NCT03903835). Top: in standard practice, a single drug is given to an unselected cohort, resulting in heterogeneous responses. Bottom: in ProBio (NCT03903835) plasma ctDNA sequencing (78-gene panel plus germline control) stratifies men with mCPRC into four predefined genomic signatures, *AR-WT*, *TP53-WT*, *TMPRSS2–ERG* fusion, *HRD*, and *TP53*-altered. Within each signature, a Bayesian algorithm adaptively randomises patients between an androgen-receptor-pathway inhibitor (ARPI) and a taxane, while a separate physician’s-choice arm remains non-adaptive. Interim data show ARPIs prolong benefit and OS in the *AR-WT* and *TP53-WT* and *TMPRSS2–ERG* groups, yield comparable outcomes to taxanes in TP53-altered disease, and are still underpowered in HRD-positive tumors. Created in BioRender. Fatima, Y. https://BioRender.com/o8msq1v (accessed on 30 June 2025). Abbreviations: ARPI, androgen-receptor-pathway inhibitor; ctDNA, circulating tumor DNA; HRD, homologous-recombination deficiency; OS, overall survival; WT, wild-type.

**Table 1 ijms-26-07475-t001:** Approved PARP-inhibitor-based therapies guided by omics testing (companion test) in advanced-stage PCa.

Combination Therapy (Dose)	Genomic Requirement (Companion Diagnostic)	Pivotal Trial (s)	Key Treated Population (*n*)	Key Efficacy Outcome	*p*-Value (Primary Endpoint †)	Notable Points
Olaparib (300 mg) vs. abiraterone/enzalutamide (monotherapy) [44,48]	*BRCA1/2* or *ATM* mutations by FoundationOne CDx, BRACAnalysis CDx or qualified liquid CDx	PROfound (phase III)	Cohort A, *n* = 245 (387 total)	rPFS 7.4 vs. 3.6 mo, HR 0.34; OS 19.1 vs. 14.7 mo, HR 0.69	<0.001	66% crossover; 33% screen-failure; grade ≥ 3 anaemia 23%
Rucaparib (600 mg) (monotherapy) [54]	Deleterious germline/somatic *BRCA1/2* (central or validated local NGS)	TRITON2 (phase II, single-arm) → TRITON3 (phase III)	TRITON2 *BRCA n* = 115 (62 RECIST-evaluable); TRITON3 *BRCA n =* 302	ORR 44% (TRITON2); rPFS 11.2 vs. 6.4 mo, HR 0.50 (TRITON3)	TRITON2—(single-arm study; no comparator, therefore no *p* value) TRITON3 < 0.0001 (rPFS); TRITON3	Accelerated approval awaiting OS verification; 47% tissue-plasma discordance; grade ≥ 3 anaemia ≈ 25%
Olaparib (300 mg) + Abiraterone (1000 mg)/Prednisone (5 mg) (combination therapy) [59]	None (all-comers); preplanned *BRCA1/2* subgroup	PROpel (phase III)	ITT *n* = 796; *BRCA* subset *n =* 85	ITT rPFS 24.8 vs. 16.6 mo, HR 0.66; *BRCA* HR 0.24; interim OS HR 0.30	<0.0001 (rPFS ITT)	Label restricted to *BRCA*; grade ≥ 3 anaemia 15–16%; 14% discontinuation; first PARP–NHAA doublet approval
Niraparib 200 mg and Abiraterone acetate (1000 mg) + prednisone/prednisolone (10 mg) (combination therapy) [62]	HRR-positive by tissue/plasma NGS; label restricted to *BRCA1/2*	MAGNITUDE (phase III)	ITT *n =* 423; *BRCA* subgroup *n =* 225	rPFS (*BRCA*) 16.6 vs. 10.9 mo, HR 0.53; HRR-pos HR 0.73	0.0014 (rPFS in *BRCA*)	OS neutral; grade ≥ 3 anaemia 29.7%; 15% discontinuation; volumetric PSMA metrics complementary
Talazoparib (0.5 mg) + Enzalutamide (160 mg) (combination therapy) [63]	HRR-deficiency mutation confirmed via FoundationOne CDx	TALAPRO-2 (phase III)	ITT *n* = 805; HRR mutation subset *n =* 399	ITT rPFS NR vs. 21.9 mo, HR 0.63; HRR subset HR 0.45	<0.0001 (rPFS ITT)	FDA approved on 20 June 2023, for HRR-mutated mCRPC. Final OS: 45.1 vs. 31.1 mo (HR 0.62; *p* = 0.0005) now on US label Grade ≥ 3 anaemia 46%; treatment discontinuation 8%. g

Abbreviations: ATM = ataxia-telangiectasia mutated; bid = bis in die (twice daily); BRCA1/2 = BReast CAncer susceptibility genes 1 and 2; CDx = companion diagnostic; HR = hazard ratio; HRR = homologous-recombination repair; ITT = intention-to-treat; mo = months; NHAA = novel hormonal-androgen-axis agent; NR = not reached; NGS = next-generation sequencing; ORR = objective response rate; OS = overall survival; PARP = poly (ADP-ribose) polymerase; Pred = prednisone/prednisolone; PSMA = prostate-specific membrane antigen; rPFS = radiographic progression-free survival; RECIST = Response Evaluation Criteria in Solid Tumors; ASCO-GU = American Society of Clinical Oncology Genitourinary Cancers Symposium; FDA = Food and Drug Administration; † Primary Endpoint: radiographic progression-free survival (rPFS).

**Table 3 ijms-26-07475-t003:** Omics-driven stratification for therapeutic interventions in clinical trials for advanced-stage prostate cancer.

Combination/Intervention (Dose)	Genomic/Molecular Requirement (Companion Assay)	Trial (Phase)	Key Treated Population (*n*)	Key Efficacy Outcome	*p*-Value (Primary End-Point)	Notable Points
Ipatasertib (400 mg) with abiraterone vs. placebo with abiraterone [84]	*PTEN*-loss by IHC (FoundationOne^®^ genomic confirmatory subset)	IPATential150 (III)	*PTEN*-loss subset ≈ 521 of 1101	rPFS 18.5 vs. 16.5 mo (HR 0.77)	0.034	OS HR 0.94; NS; grade ≥ 3 AEs 70% vs. 39%
Intermittent docetaxel (withheld when plasma *mGSTP1* clears) vs. continuous docetaxel (75 mg m^−2^ q3w) [85]	Detectable *mGSTP1* ctDNA at baseline and clearance after 2 cycles (mSTRAT assay)	GUIDE (II)	Target 28 (was 120); 6 randomised to date	Ongoing—primary rPFS	NS	First epigenetic ctDNA-adaptive chemotherapy; slow accrual, trial now closed to recruitment
Evolocumab (420 mg sc q4w) + SOC therapy [88]	PCPro lipidomic score > −1.1903 (5-analyte ceramide/TRG/CH total)	Evolocumab-PCPro (II, single-arm)	PCPro-positive pts (target ≈ 40)	Primary: PCPro re-classification at wk 12 (ongoing)	NS	Precision metabolomics strategy; endpoints include PSA_50_ and broad lipidomic shifts
Nivolumab and ipilimumab (two schedules) [89]	Immunogenic signature (ImS^+^): MMR-d, DDR-d or high TILs ≥ 20%	NEPTUNES (II)	C1 35, C2 36	Composite response rate 40% vs. 25% (overall 32%)	(≥40% predefined as meaningful)	Higher grade 3–4 AEs in IPI-intense arm; responders enriched for MMR-d & *BRCA*/*ATM* loss
AZD4635 75 mg qd ± durvalumab 1500 mg q4w [90]	Baseline 14-gene AdenoSig (exploratory, not selection)	NCT02740985 (Ia/b)	108 total (mono 65; combo 43)	ORR 5% mono, 16% combo; PSA_50_ 22% combo; AdenoSig-high rPFS 21 vs. 8.7 wks (HR ≈ 0.46)	NS	First-in-human adenosinergic blockade; benefit confined to AdenoSig-high tumors
Early docetaxel 75 mg m^−2^ q3w + ADT vs. ADT alone [91]	PAM50 luminal-basal subtype & Decipher GC (RNA)	CHAARTED correlative (III)	Analytic set *n* = 160	Luminal B OS benefit HR 0.45 (*p* = 0.007); GC-high greatest 3-yr OS gain (+25%)	0.007 (luminal B)	Transcriptomics refines who benefits from upfront docetaxel in mHSPC
Outcome-adaptive ARPI (abiraterone/enzalutamide) vs. taxane (docetaxel/cabazitaxel) [92]	Real-time 78-gene ctDNA panel → 5 molecular signatures (AR–/TP53-WT, TP53-alt, TMPRSS2-ERG, HRD, unselected)	ProBio platform (adaptive RCT)	First 218 randomisations (193 pts)	ARPI vs. taxane: TTNLCB† 11.1 vs. 6.9 mo (STR 1.60); OS 38.7 vs. 21.7 mo	Bayesian adaptive (no fixed α)	Greatest ARPI benefit in AR–/TP53-WT & TMPRSS2-ERG; proof-of-concept for same-day ctDNA-guided randomisation

Abbreviations: ARPI = androgen-receptor-pathway inhibitor; ITT = intention-to-treat; mGSTP1 = methylated-GSTP1; mo = months; NE = not estimable; NR = not reached; ORR = objective-response rate; OS = overall survival; PSA_50_ = ≥50% PSA decline; rPFS = radiographic progression-free survival; STR = survival-time ratio; NS = non-significant.

**Table 4 ijms-26-07475-t004:** Summary of actionable biomarkers and corresponding therapeutic options.

Biomarker (Analytical Platform/Biospecimen)	Registration/Approval or Investigational Status (as of 2025)	Matched Therapeutic Scheme
BRCA1/2 pathogenic mutation (NGS/tissue or plasma) [51,56,63]	Approved—FDA-cleared companion tests (FoundationOne CDx, BRACAnalysis)	PARP inhibitors: olaparib with/without abiraterone, rucaparib niraparib combined with abiraterone, talazoparib combined with enzalutamide
High PSMA expression on ^68Ga-PSMA-11 PET/CT test (molecular imaging) [71]	Approved-Phase III (VISION), imaging-based test as a companion to therapy	Radioligand therapy lutetium-177 vipivotide tetraxetan (Pluvicto™)
*PTEN*-loss (IHC or NGS assay/blood plasma samples) [84]	Phase III (IPATential150)	Ipatasertib 400 mg daily plus abiraterone
PCPro ceramide lipidomic score (high) (Ceramide biomarker panels/Blood plasma sample) [88]	Phase II (PCPro)	Evolocumab (PCSK9 inhibitor) added to standard therapy
Immunogenic signature assay (ImS^+^): MMR-d, DDR-d or high TILs ≥ 20%, (Blood specimens; plasma or serum) [89]	Phase II (NEPTUNES; ongoing)	Nivolumab (PD-1 inhibitor) in combination with ipilimumab (CTLA4 inhibitor)

## Data Availability

No new data were created or analyzed in this study. Data sharing does not apply to this article.

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
