# Peer review of "Omics-Mediated Treatment for Advanced Prostate Cancer: Moving Towards Precision Oncology"

_ijms, 2025, doi:10.3390/ijms26157475_

Round 1
Reviewer 1 Report
Comments and Suggestions for Authors
This manuscript tackles an important and timely topic in the management of advanced prostate cancer, focusing on omics-guided approaches to therapy. The structure is logical, the scope is broad, and the references are extensive. At first glance, the manuscript gives the impression of being comprehensive.
However, many sections would benefit from deeper analysis and more critical synthesis. While the authors cite a large number of clinical trials and biomarkers, the manuscript leans heavily toward descriptive listing. There is less discussion than expected around conflicting findings, limitations of current approaches, or the real-world translational challenges that clinicians and researchers face in applying these omics strategies. The overall analytical depth of the manuscript could be strengthened to better match its breadth.
This is especially noticeable in the PSMA-directed therapy section, which is comparatively brief and omits several clinically relevant complexities. I recommend adding a short paragraph discussing the biological determinants of PSMA expression and heterogeneity, including spatial and temporal variation. The current version does not address important issues such as:
-
The impact of neuroendocrine differentiation on PSMA expression
-
Lineage plasticity and its role in low or absent PSMA expression in treatment-resistant tumors
-
The regulatory mechanisms that drive PSMA heterogeneity across AR-positive and AR-negative metastatic disease
In addition, the manuscript would benefit from incorporating recent developments in emerging imaging and biomarker strategies that may help address these challenges. For instance, several recent studies have examined the differential expression of glucose transporters and hexokinases in prostate cancer with a neuroendocrine gene signature, providing a mechanistic rationale for integrating ^18F-FDG PET in PSMA-low tumors. This perspective also strengthens the conceptual link between radiomics and metabolomics, which the manuscript highlights as an emerging frontier.
In summary, while this is a timely and well-structured manuscript, I recommend acceptance with minor revisions, contingent on expanding the PSMA section to address the issues outlined above and improve the overall depth of analysis.
Author Response
Replies to Reviewers’ Comments
Reviewer 1 – Comments
This manuscript tackles an important and timely topic in the management of advanced prostate cancer, focusing on omics-guided approaches to therapy. The structure is logical, the scope is broad, and the references are extensive. At first glance, the manuscript gives the impression of being comprehensive.
Reviewer 1 –Response
We sincerely thank the Reviewer for the thoughtful and detailed evaluation of our manuscript and for the specific suggestions that have helped us clarify and improve the quality of our work, particularly in providing critical statements in this evolving field. Below we provide our point-by-point replies to each comment, along with a summary of the changes made to the revised version (highlighted in yellow in the manuscript and given below as a copy-pasted text from the manuscript with the section, page, and line number):
Reviewer 1 – Comment 1:
However, many sections would benefit from deeper analysis and more critical synthesis. While the authors cite a large number of clinical trials and biomarkers, the manuscript leans heavily toward descriptive listing. There is less discussion than expected around conflicting findings, limitations of current approaches, or the real-world translational challenges that clinicians and researchers face in applying these omics strategies. The overall analytical depth of the manuscript could be strengthened to better match its breadth.
Reviewer 1 – Response to comment 1:
To address this comment, we added various sections about the points mentioned above, throughout the article, by essentially adding a conclusive paragraph after each Section: Section 1 on summarising the existing therapies that are guided based on predictive/companion markers either as monotherapy or in combination therapy, Section 2 on PSMA-directed therapies, and Section 3 referring to investigational therapies guided by omics markers to stratify patients when entering clinical trials.
Along these lines, an extensive part of the Discussion is now enriched with our critical views on:
- Technical considerations related to failure of omics-based screening (especially related to NGS and liquid biopsies).
- The limitations related to the screening of archival tissues and their moderate concordance in terms of mutational status, in comparison with liquid biopsy genomics assays.
- Challenges related to variability in PSMA expression.
- High rate of adverse events for combination treatments (even after omics stratification) and the fact that predictive markers cannot currently predict side effects,
- Last but not least, other considerations that hinder broad adoption of omics-based assays, among others, cost, time to treat window, and lack of standardized SOPs, are now presented.
The copied and pasted sections discussing the above resolved comments are given below with section, page, and line number.
Section 3.1.1. PARP inhibitors, page 8, lines 278-283:
“While genomics testing for the presence of HRR mutations is a prerequisite for receiving PARP inhibitors, as evident based on the above clinical trial data, several technical limitations exist. A high percentage (approximately 25 – 35 %) of patients fail the screening, due to sub-optimal (for genomics testing) archival tissue quality and/or disconcordance of mutation status between peripheral plasma and tissue (up to 80% depending on the targeted mutations)[57,58]”
Section 3.1.2. Combinatorial therapies, page 10, lines 352-370:
“As in the targeted monotherapies, also in the combinatorial setting, routine multi-gene NGS of both tissue and circulating tumor DNA is now standard of care, with the 2025 NCCN guideline mandating HRR-gene screening (BRCA1/2, ATM, PALB2, etc.) for every mCRPC patient[65]. Clinical applicability is direct, as the FDA’s 2023 approval of the combination of olaparib plus abiraterone/prednisone for BRCA-mutated mCRPC patients is tied to a companion diagnostic test such as, FoundationOne CDx and FoundationOne Liquid CDx, that can be run on archival tissue, or plasma[66]. Similarly, in 2024, the fixed-dose niraparib/abiraterone tablet (Akeega) was approved together with NGS screening with FoundationOne Liquid CDx as a companion test. Yet, despite these landmark advances, several limitations and uncertainties complicate the integration of PARP inhibitors into the routine management of mCRPC. First of all, conflicting clinical findings highlight that not all HRR gene mutations confer equal sensitivity, with BRCA2-mutant tumors achieving significantly better outcomes than BRCA1 or other HRR-altered subtypes[67]. NGS screening approaches also come with technical limitations as tissue-based NGS screening frequently fails due to inadequate tumor material, and plasma-based assays show only modest concordance with the tissue mutational status. Additionally, combination therapies with PARP inhibitors (PARPi), especially with chemotherapy or other DNA-damaging agents, result in increased toxicity, mainly hematologic effects (anemia, neutropenia, thrombocytopenia) which often exceed those seen with monotherapy[68].”
Section 3.1.3 Imaging-based PSMA (prostate-specific membrane antigen) directed radioligand therapy, page 13
Limitations of the VISION trial: (line 398-401)
“Notably, VISION excluded patients who had any PSMA-negative lesion measuring ≥ 2.5 cm in lymph nodes or ≥ 1 cm in visceral or bone sites, meaning the OS benefit cannot be extrapolated to tumors with mixed or low PSMA expression[71].”
Limitations of PSMAfore trial: (line 408- 410)
“Despite the impressive rPFS hazard ratio, 8 % of screened patients were ineligible due to discordant PSMA uptake, underscoring the need for back-up imaging in heterogeneous disease.”
Discussion (pages 31, 32, 33, 34), (lines 1017- 1139)
“This review compiles evidence demonstrating that molecular stratification is no longer an experimental addition in advanced PCa but rather a prerequisite for optimal therapy selection. PARP inhibitors, either as monotherapy or in combinatorial therapies, demonstrate benefit within BRCA1/2- and/or HRR mutation-bearing tumors[56,59,60]. Moreover, PSMA-directed radioligand therapy with ^177Lu-PSMA-617(VISION) is another example of FDA-approved biomarker-guided therapies. Additionally, PARP blockade is the second FDA-approved, companion diagnostic-mandated option to independently predict survival[71]. A summary of the most promising biomarkers associated with therapeutic pharmacological intervention, either already approved or in running clinical trials, is given in Table 4.
At the same time, multiple reports shed light on omics-based biomarkers that are applied as stratification means within running clinical trials (summarized in the second section of this article and Table 3). An exemplary case of an adaptive clinical trial where omics biomarkers are guiding treatment is the ProBio trial, where ctDNA biomarker-based screening is used to randomize patients with advanced PCa into four treatment arms, with the possibility to adapt therapy based on the biomarker status[92].
Despite the evident progress in the field, as summarized also in Table 4, there are still several challenges that hinder the wide applicability of omics-mediated therapeutic schemes: i) on the one hand related to the heterogenic nature of advanced PCa, ii) secondly, associated with technical limitations of the biomarker testing and iii) lastly, with inherent limitations of targeted (and even more combination thereof) therapies.
Firstly, mCRPC presents with intratumor heterogeneity, whereby molecularly distinct subclones coexist within the same tumor, or across metastatic sites. This spatial and temporal heterogeneity contributes to significant variability, particularly in the mutational status when comparing primary to metastatic sites[127]. This becomes a major clinical challenge when treating patients with advanced metastatic disease, based on a mutation screening that was performed in primary and/or archival tissue blocks. Depending on the targeted mutation, concordance varies relatively high, though not absolute, concordance exists between primary prostate tumors and metastatic sites for pathogenic DDR alterations. In the largest paired-sample study to date, 84% of DDR mutations detected in metastatic tissue or ctDNA were already present in the corresponding archival primary specimen (95% CI 71–92%), indicating that most are truncal events acquired early in tumor evolution[58]. Real-world datasets confirm this, showing moderate to substantial agreement between primary–ctDNA (κ = 0.59) and metastatic–ctDNA (κ = 0.65) pairs, and suggesting that specimens are largely interchangeable for actionable DDR variants[128]. Archival primary tissue is an appropriate first-line specimen for DDR and mismatch-repair (MMR) testing in mCPRC. However, negative, outdated, or equivocal results should trigger reflex testing of a contemporary metastatic biopsy or high-tumor-fraction ctDNA, particularly at progression or after exposure to targeted agents such as PARP inhibitors[129]. Nevertheless, therapy-induced variants, such as multiclonal BRCA2 reversion mutations or acquired AR alterations, have been identified exclusively in post-treatment metastatic biopsies or ctDNA in up to one-third of advanced PCa cases, reflecting both spatial and temporal heterogeneity. Reliance solely on the primary tumor in such contexts, therefore, risks missing emergent resistance mechanisms and may lead to unexpected treatment resistance[130]. To address such intra-patient heterogeneity in mCRPC, emerging strategies such as serial liquid biopsies, multiregional sampling, and longitudinal genomic profiling are applied to capture clonal evolution and resistance mechanisms more effectively[127,131]. These approaches help overcome sampling bias and improve the detection of subclonal mutations, allowing for more accurate patient selection and adaptive treatment strategies[132]. However, challenges remain in standardizing these methods and integrating them into routine clinical workflows.
An additional challenge of the omics-guided treatment approaches is associated with archival tissue analysis (particularly concerning NGS). Archival tissue blocks are frequently characterized by low tumor content and degradation of the biological material. The importance of high-quality BRCA1/2 testing for guiding treatment with PARP inhibitor was underscored within a recent multicenter study[133]. Sequencing success in 954 prostate‑cancer FFPE specimens fell from 88% in blocks < 1 year old to 69% when storage exceeded two years, with DNA concentration and fragmentation emerging as independent predictors of failure. These real-world data support policies that prioritize fresh biopsies or rapid testing to avoid missing PARP-eligible patients[133].
Liquid-based biopsy, as one of the proposed solutions to the above problem, is promising, but it is still evolving with its routine use in broader clinical practice to await further validation. Liquid biopsy-based markers are incorporated into clinical trials for advanced PCa to assess eligibility for specific therapies, monitor treatment response, assess minimal residual disease (MRD), and assess response to PARPi or androgen receptor signaling inhibitors[134]. Moreover, blood or serum-based liquid biopsies are generally favored for their ease of sampling and ability to capture disease heterogeneity[135]. They are used as a complement to traditional biopsy, especially when tissue is difficult to access or not available[135]. One important factor is that currently, concordance between tissue and liquid biopsy in detecting HRR and other actionable mutations is moderate and context-dependent, as mentioned above, largely depending on the mutation type. Such moderate concordance underscores the complementary nature of these approaches rather than their interchangeability. Due to this, liquid biopsy is often considered a useful adjunct but not a standalone replacement for tissue testing in many clinical scenarios[136].
Importantly, cost-effectiveness of targeted treatment approaches, exemplified by PARP inhibitors and PSMA-targeted radioligand therapy, presents a complex and often conentious landscape. For example, PSMA radioligand therapy meets Germany’s willingness-to-pay threshold at about €69,000 per quality-adjusted life year (QALY) but falls short in countries with stricter benchmarks based on models where only health-system costs are considered. When patient out-of-pocket expenses and productivity losses are added (“societal perspective”), even PSMA therapy often fails to demonstrate acceptable value in many high-income settings[137]. PARP inhibitors face even steeper economic hurdles, as their high cost result in incremental cost-effectiveness ratios above conventional cutoffs, approximately CAD$565,000/QALY in Canada and AU$144,000/QALY in Australia[138,139]. Within this context, treatment with a PARP inhibitor such as olaparib for a year can cost between US$100,000 and US$150,000 in the United States and around €5,000 monthly (approximately €60,000 yearly) in Germany, underscoring the financial challenges associated with accessing these therapies.
Despite these challenges, the field is evolving with already successful paradigms paving the way. Certainly, omics screens offer unique possibilities because of the complex and high-dimensional nature of the data (profiles). As omics data evolve, we envision that integration of multiple omics datasets, and/or molecular signatures (networks or pathways) will improve the prediction of therapy response, stratify patients more accurately, and subsequently more accurately guide therapy. We would also expect that an improved understanding of the molecular pathophysiology of the advanced PCa (through multiple molecular layers) can fill an existing clinical need for predicting side effects, which are particularly pronounced after patients receive a combination of targeted therapies. At the same time, this requires standardization and alignment of the standard operating protocols and laboratory procedures, and characterization of the variability that can be introduced based on the measurements in different analytical platforms, along with validation of analytical performance according to regulatory requirements [140]. In parallel, harmonization of heterogeneous data formats, normalization methods, and annotation standards[141] are becoming a prerequisite for the implementation of such multi-omics approach.
Besides the advancements at the molecular level, AI-based tools are also increasingly used: First, multimodal large‑language‑models(LLM) that can incorporate a variety of modalities, including across pathology slides, imaging, genomics, and by guidelines, now achieve up to 80% decision accuracy in simulated tumor‑board cases, vastly outperforming standalone LLMs[142]. Second, early studies within collaborative European cancer research network such as EUCAIM (EUropean Federation for CAncer Images), OPTIMA(Optimal Treatment for Early-stage Prostate Cancer) and other net-works suggest that federated models(machine‑learning systems trained across multiple sites, pooling the raw data in one place) match or modestly exceed public, centrally trained models; recent demonstrations involve >60 institutions, but further large-scale validation is still in progress[143]. Third, AI‑driven feature extraction radiomics, pathomics, and longitudinal ctDNA‐omics already predict immunotherapy response with up to 89 % (AUC 0.98–1.00) accuracy and are being embedded in adaptive trial de-signs[144]. Moving forward, complementing existing therapeutic approaches with the above multi-parametric models could lead to true precision oncology and improved treatment of advanced PCa.
- Outlook
Our synthesis integrates five omics dimensions and explicitly links them to therapy development stages, thereby providing the first complete overview that spans approved drugs, investigational combinations, and computational repurposing. Previous narrative reviews have typically focused on a single omics layer, genomics-only surveys of PARP inhibition, or meta-analyses of PSMA radioligand therapy. The summarized results are based on a search confined to Web of Science and publications from 2021 onward, so earlier or non-indexed studies may have been missed. Additionally, the analyses relied on aggregate study-level data, preventing formal examination of within-study heterogeneity or predefined subgroup effects, while the risk-of-bias ratings were based solely on information reported in each study. Under these considerations, this article provides a comprehensive view on where we stand in terms of omics-guided therapies in advanced PCa, as well as a perspective on a “test-first-treat-later” paradigm. Both upfront germline and somatic DNA-repair profiling and PSMA-PET/CT imaging are included in the latest ASCO guidelines for men with mCPRC disease[145]. Serial liquid biopsies add a dynamic dimension, allowing early intensification or therapy switching rather than waiting for reaching endpoints like PFS[135]. Importantly, several studies show that prognostic assays need not be predictive: CTC enumeration and ctDNA fraction independently stratify risk even when they do not direct treatment choice, providing a clear numerical reference point for shared decision-making[135,136]. Taking this together, despite the technical and analytical challenges, the field is rapidly expanding, with multiple omics layers expected to improve our understanding of underlying molecular pathology and thus improve therapeutic guidance in disease stages that are still difficult to treat.”
Reviewer 1 – Comment 2:
This is especially noticeable in the PSMA-directed therapy section, which is comparatively brief and omits several clinically relevant complexities. I recommend adding a short paragraph discussing the biological determinants of PSMA expression and heterogeneity, including spatial and temporal variation. The current version does not address important issues such as:
- The impact of neuroendocrine differentiation on PSMA expression
- Lineage plasticity and its role in low or absent PSMA expression in treatment-resistant tumors
- The regulatory mechanisms that drive PSMA heterogeneity across AR-positive and AR-negative metastatic disease
In addition, the manuscript would benefit from incorporating recent developments in emerging imaging and biomarker strategies that may help address these challenges. For instance, several recent studies have examined the differential expression of glucose transporters and hexokinases in prostate cancer with a neuroendocrine gene signature, providing a mechanistic rationale for integrating ^18F-FDG PET in PSMA-low tumors. This perspective also strengthens the conceptual link between radiomics and metabolomics, which the manuscript highlights as an emerging frontier.
In summary, while this is a timely and well-structured manuscript, I recommend acceptance with minor revisions, contingent on expanding the PSMA section to address the issues outlined above and improving the overall depth of analysis.
Reviewer 1 – Response to comment 2:
To address the Reviewer’s comments, extensive revisions have been made in the PSMA section: a), Some background information on the PSMA is now provided, and the already existing trial examples PSMAfore and VISION have been highlighted. b), the AR-tumor impact and the role of lineage plasticity are also discussed, and its impact on the ^177Lu-PSMA-617 radionuclide therapy is highlighted. The important implications for Clinicians have also been highlighted in the case of PSMA flare. The regulatory mechanisms that drive PSMA heterogeneity are now covered as well. Additionally, the alternative imaging strategies, like 18F-FDG PET in the case of very high-risk PCa patients, are now presented, whereas at the end of the section, a brief overview of the alternative therapies is provided. The respective sections are pasted below:
Section 3.1.3: Imaging-based PSMA (prostate-specific membrane antigen) directed radioligand therapy, pages 13, 14, and 15.
Line number (379-386): “PSMA is a transmembrane folate hydrolase enzyme that was found with increased protein expression levels in PCa[69]. As such, this protein has served as a cell-surface imaging biomarker and therapeutic target in advanced PCa. Nevertheless, as PSMA expression is often suppressed in AR-negative advanced prostate cancers and liver metastases, a definition of PSMA positivity is a prerequisite for the use of PSMA-radioligand therapy; in addition, PSMA heterogeneity across tumoral and spatial, inter-patient, intra-patient, lineage, and temporal dimensions presents a challenge for efficient patient management[69,70].“
Line number (430-484): “Importantly, as PSMA expression is suppressed by AR, AR-positive tumors typically exhibit low PSMA expressions. Upon AR pathway inhibition, there may be a consequent up‑regulation of PSMA; however, if the tumor undergoes neuroendocrine differentiation(NED), driven by factors like N‑Myc and EZH2, epigenetic silencing of FOLH1 results in PSMA‑low or ‑negative subclones[75]. At the same time, lineage plasticity is the ability of tumor cells to shift between luminal, neuroendocrine, and other cell phenotypes. It not only enables survival under therapeutic pressure but also fosters the emergence of treatment-resistant phenotypes characterized by low or absent PSMA expression[76]. PSMA negativity then predicts poor response to ^177Lu-PSMA-617 radioligand therapy. Transient “PSMA flare” after brief AR inhibition shortly after starting AR inhibition therapy (ADT or AR blockers) can increase PSMA target density and may be exploited to enhance the efficacy of PSMA-directed therapies. Short-term androgen blockade (termed STAB) has been shown to transiently upregulate PSMA ex-pression in PCa, resulting in SUVₘₐₓ increases of approximately 30–50% on PSMA PET imaging. Integration of STAB into imaging or treatment protocols may improve diagnostic sensitivity and therapeutic efficacy, particularly in cases where baseline PSMA expression is suboptimal. Nevertheless, further prospective studies are warranted to standardize timing and patient selection for this sensitization approach[77]. Recognizing dynamic changes in PSMA expression is clinically critical for imaging interpretation, therapy selection, and realizing the potential of combination or sequential therapies.
The clinical detectability and quantification of PSMA expressions in PCa lesions are directly influenced by the choice of radiotracer and imaging technology. Although [^68Ga]-labeled agents have traditionally been the clinical standard, next-generation fluorinated compounds like [^18F]DCFPyL offer advantages, including improved image resolution and delayed imaging protocols (referring to acquisition of image scans at 90–120 minutes, or even up to 3 hours post-injection, rather than the standard image ac-quisition within an hour), which have demonstrated superior sensitivity for local and nodal staging in high- and very high-risk patients. For example, a recent study demonstrated that [^18F]DCFPyL PET/CT significantly improves the detection of loco-regional disease in patients with advanced PCa who are candidates for radical therapy, ensuring more accurate risk stratification, surgical planning, and more effective prioritization for PSMA-targeted therapies[78].
Moreover, in AR‑negative double‑negative prostate cancer (DNPC), PSMA expression can be virtually absent despite high metabolic activity, whereas AR‑positive CRPC often retains heterogeneous PSMA uptake[75]. These resistant subpopulations, often AR-negative or double-negative, display marked molecular and metabolic divergence from classical adenocarcinoma, including upregulation of glucose transporters and hexokinases that facilitate increased glycolytic activity. Recent papers show PSMA-low tumors overexpress sugar transporters (GLUT1) and hexokinases, increasing the uptake in an FDG (glucose) PET scan, even when PSMA-PET signal is low. This metabolic shift provides a mechanistic rationale for integrating ^18F-FDG PET in the diagnostic workflow for PSMA-low or neuroendocrine tumors, complementing ^68Ga-PSMA PET [79]. In one case report, FDG PET/CT identified lesions not seen on PSMA PET/CT, underscoring the potential for FDG-positive and PSMA-negative discordance[80]. This means that while PSMA-PET/CT did not show any activity in these areas, FDG-PET/CT identified lesions suggestive of active disease. Excluding patients with FDG-positive but PSMA-negative disease from PSMA-targeted RLT(Radioligand therapy) improves response rates and spares them from an ineffective treatment, and should be triaged to alternative RLT targets such as GRPR or DLL3[81–83].
Recent studies further highlight the promise of advanced biomarker approaches, such as radiomic and metabolomic profiling (discussed in Section 3.5 of this review), which leverage high-dimensional imaging and molecular data to capture clinically relevant disease heterogeneity that is not apparent with standard imaging alone. These mechanistic insights explain both the variable responses observed in PSMA‑targeted radioligand trials.”

Reviewer 2 Report
Comments and Suggestions for Authors
Authors have to be commended for the timely and relevant overview of how omics technologies can contribute to more precise and individualized treatments in advanced prostate cancer. Precision medicine with targeted therapies represents the best treatment for prostate cancer not suitable for surgical therapy.
Thus, the topic is clearly important, and the authors succeed in outlining the biological rationale behind omics-based approaches.
Major Comments
While the scientific background is well developed, the paper would benefit from a deeper discussion of how omics-guided strategies are currently applied in real-world clinical practice. Specific examples (e.g., NGS panels, liquid biopsy use in metastatic settings) would increase the translational value. Consider citing: DOI 10.3390/cancers17101705.
A summary table could help clarify actionable biomarkers and corresponding therapeutic options.
The discussion should focus more on:
- How do we overcome intra-patient heterogeneity?
- What are the current limitations in bioinformatics standardization and omics data interpretation?
- Cost-effectiveness and accessibility of these approaches in routine practice.
Including recent or ongoing clinical trials linking omics stratification to drug response would add relevance. A brief section on anticipated future developments as AI-enhanced analysis would also help contextualize the evolution of precision oncology.
Minor Comments
Ensure consistent terminology (e.g., switch between “castration-resistant prostate cancer” and “CRPC”).
Recheck some grammatical constructions (e.g., long and passive sentences could be simplified for clarity).
Please cite the following papers about mCRPC and PSMA PET:
- DOI: 1007/s00259-025-07133-1
- DOI: 10.1002/advs.202305724
- DOI: 3390/cancers15194809
Author Response
Reviewer 2 – Comments
Authors have to be commended for the timely and relevant overview of how omics technologies can contribute to more precise and individualized treatments in advanced prostate cancer. Precision medicine with targeted therapies represents the best treatment for prostate cancer not suitable for surgical therapy.
Thus, the topic is clearly important, and the authors succeed in outlining the biological rationale behind omics-based approaches.
Reviewer 2 –Response
We sincerely thank the Reviewer for the positive feedback. The constructive and thoughtful comments helped to significantly improve the review paper. Below we provide our point-by-point replies to each comment, along with a summary of the changes made to the revised version (highlighted in yellow in the manuscript and given below as a copy-pasted text from the manuscript with the section, page, and line number):
Reviewer 2 – Major Comments
Reviewer 2 – Comment 1
While the scientific background is well developed, the paper would benefit from a deeper discussion of how omics-guided strategies are currently applied in real-world clinical practice. Specific examples (e.g., NGS panels, liquid biopsy use in metastatic settings) would increase the translational value.
Reviewer 2 – Response to Comment 1
To address this comment, we have added a paragraph about the implementation of Omics-guided strategies in the real world. Moreover, extensive technical considerations related to the failure of omics-based screening (especially related to NGS) have also been presented in the discussion. Additionally, we also discuss the importance of Spatial and Temporal Heterogeneity and their clinical implications, as well as the challenges associated with the use of liquid biopsies(discussion section). The page and line number discussing the above sections are given below:
Pasted text from the manuscript- 1st part (pg. 32 and line number 1047-1065):
“Depending on the targeted mutation, concordance varies relatively high, though not absolute, concordance exists between primary prostate tumors and metastatic sites for pathogenic DDR alterations. In the largest paired-sample study to date, 84% of DDR mutations detected in metastatic tissue or ctDNA were already present in the corresponding archival primary specimen (95% CI 71–92%), indicating that most are truncal events acquired early in tumor evolution[58]. Real-world datasets confirm this, showing moderate to substantial agreement between primary–ctDNA (κ = 0.59) and metastatic–ctDNA (κ = 0.65) pairs, and suggesting that specimens are largely interchangeable for actionable DDR variants[128]. Archival primary tissue is an appropriate first-line specimen for DDR and mismatch-repair (MMR) testing in mCPRC. However, negative, outdated, or equivocal results should trigger reflex testing of a contemporary metastatic biopsy or high-tumor-fraction ctDNA, particularly at progression or after exposure to targeted agents such as PARP inhibitors[129]. Nevertheless, therapy-induced variants, such as multiclonal BRCA2 reversion mutations or acquired AR alterations, have been identified exclusively in post-treatment metastatic biopsies or ctDNA in up to one-third of advanced PCa cases, reflecting both spatial and temporal heterogeneity. Reliance solely on the primary tumor in such contexts, therefore, risks missing emergent resistance mechanisms and may lead to unexpected treatment resistance[130].”
Pasted text from the manuscript- 2nd part (pg. 32 and 33 and line numbers 1081-1094):
“Liquid-based biopsy, as one of the proposed solutions to the above problem, is promising, but it is still evolving with its routine use in broader clinical practice to await further validation. Liquid biopsy-based markers are incorporated into clinical trials for advanced PCa to assess eligibility for specific therapies, monitor treatment response, assess minimal residual disease (MRD), and assess response to PARPi or androgen receptor signaling inhibitors[134]. Moreover, blood or serum-based liquid biopsies are generally favored for their ease of sampling and ability to capture disease heterogeneity[135]. They are used as a complement to traditional biopsy, especially when tissue is difficult to access or not available[135]. One important factor is that currently, concordance between tissue and liquid biopsy in detecting HRR and other actionable mutations is moderate and context-dependent, as mentioned above, largely depending on the mutation type. Such moderate concordance underscores the complementary nature of these approaches rather than their interchangeability. Due to this, liquid biopsy is often considered a useful adjunct but not a standalone replacement for tissue testing in many clinical scenarios[136].”
Reviewer 2 – Comment 2
Consider citing: DOI 10.3390/cancers17101705.
Reviewer 2 – Response to Comment 2
We thank the Reviewer for the valuable suggestion. This article has now been added accordingly on page 32 and lines 1072-1080, and is pasted below:
“An additional challenge of the omics-guided treatment approaches is associated with archival tissue analysis (particularly concerning NGS). Archival tissue blocks are frequently characterized by low tumor content and degradation of the biological material. The importance of high-quality BRCA1/2 testing for guiding treatment with PARP inhibitor was underscored within a recent multicenter study[133]. Sequencing success in 954 prostate‑cancer FFPE specimens fell from 88% in blocks < 1 year old to 69% when storage exceeded two years, with DNA concentration and fragmentation emerging as independent predictors of failure. These real-world data support policies that prioritize fresh biopsies or rapid testing to avoid missing PARP-eligible patients[133].”
Reviewer 2 – Comment 3
A summary table could help clarify actionable biomarkers and corresponding therapeutic options.
Reviewer 2 – Response to Comment 3
In response to Reviewer’s comments, the following table on the biomarkers and the therapeutic options, either in approved or in promising clinical trials, has been added to the manuscript on page 31, lines 1029-1030, and it is pasted below:
Biomarker (analytical platform/ biospecimen) |
Registration/ Approval or Investigational Status (as of 2025) |
Matched therapeutic scheme |
BRCA1/2 pathogenic mutation (NGS/ tissue or plasma) [51,56,63] |
Approved –FDA-cleared companion tests (FoundationOne CDx, BRACAnalysis) |
PARP inhibitors: olaparib with/ without abiraterone, rucaparib niraparib combined with abiraterone, talazoparib combined with enzalutamide |
High PSMA expression on ^68Ga‑PSMA‑11 PET/CT test (molecular imaging) [71] |
Approved-Phase III (VISION), imaging-based test as a companion to therapy |
Radioligand therapy lutetium‑177 vipivotide tetraxetan (Pluvicto™) |
PTEN-loss (IHC or NGS assay/ blood plasma samples) [84] |
Phase III (IPATential150) |
Ipatasertib 400 mg daily plus abiraterone |
PCPro ceramide lipidomic score (high) (Ceramide biomarker panels/ Blood plasma sample) [88] |
Phase II (PCPro) |
Evolocumab (PCSK9 inhibitor) added to standard therapy |
Immunogenic signature assay (ImS⁺): MMR-d, DDR-d or high TILs ≥ 20 %, (Blood specimens; plasma or serum) [89] |
Phase II (NEPTUNES; ongoing) |
Nivolumab (PD-1 inhibitor) in combination with ipilimumab (CTLA4 inhibitor) |
Reviewer 2 – Comment 4
The discussion should focus more on:
How do we overcome intra-patient heterogeneity?
What are the current limitations in bioinformatics standardization and omics data interpretation?
Cost-effectiveness and accessibility of these approaches in routine practice.
Reviewer 2 – Response to Comment 4
How do we overcome intra-patient heterogeneity
To address the comment, the reason behind intra-patient heterogeneity has been added now, as stated in the discussion section, and the possible solution to overcome it has also been provided along with it and the respective text is pasted below from page 32:
Pasted text from the manuscript- 1st part (Lines 1037-1047): “Despite the evident progress in the field, as summarized also in Table 4, there are still several challenges that hinder the wide applicability of omics-mediated therapeutic schemes: i) on the one hand related to the heterogenic nature of advanced PCa, ii) secondly, associated with technical limitations of the biomarker testing and iii) lastly, with inherent limitations of targeted (and even more combination thereof) therapies.
Firstly, mCRPC presents with intratumor heterogeneity, whereby molecularly distinct subclones coexist within the same tumor, or across metastatic sites. This spatial and temporal heterogeneity contributes to significant variability, particularly in the mutational status when comparing primary to metastatic sites[127]. This becomes a major clinical challenge when treating patients with advanced metastatic disease, based on a mutation screening that was performed in primary and/or archival tissue blocks.”
Pasted text from the manuscript- 2nd part (Lines 1065- 1071): To address such intra-patient heterogeneity in mCRPC, emerging strategies such as serial liquid biopsies, multiregional sampling, and longitudinal genomic profiling are applied to capture clonal evolution and resistance mechanisms more effectively[127,131]. These approaches help overcome sampling bias and improve the detection of subclonal mutations, allowing for more accurate patient selection and adaptive treatment strategies[132]. However, challenges remain in standardizing these methods and integrating them into routine clinical workflows.
What are the current limitations in bioinformatics standardization and omics data interpretation?
There are several issues involved, with major ones being differences in the laboratory procedures, differences in data generation platforms, and inconsistency in data annotations, which collectively prohibit integration of multi-dimensional omics data and comparison. The following section has been added in the discussion section on page 33 and lines 1109- 1124, and it reads as follows:
“Despite these challenges, the field is evolving with already successful paradigms paving the way. Certainly, omics screens offer unique possibilities because of the complex and high-dimensional nature of the data (profiles). As omics data evolve, we envision that integration of multiple omics datasets, and/or molecular signatures (networks or pathways) will improve the prediction of therapy response, stratify patients more accurately, and subsequently more accurately guide therapy. We would also expect that an improved understanding of the molecular pathophysiology of the advanced PCa (through multiple molecular layers) can fill an existing clinical need for predicting side effects, which are particularly pronounced after patients receive a combination of targeted therapies. At the same time, this requires standardization and alignment of the standard operating protocols and laboratory procedures, and characterization of the variability that can be introduced based on the measurements in different analytical platforms, along with validation of analytical performance according to regulatory requirements [140]. In parallel, harmonization of heterogeneous data formats, normalization methods, and annotation standards[141] is becoming a pre-requisite for the implementation of such multiomics approach.”
Cost-effectiveness and accessibility of these approaches in routine practice.
We have expanded the “Cost‑effectiveness and accessibility” paragraph to incorporate respective information on approved treatments, such as PARP inhibitors and PSMA-directed radioligand therapy, and did the comparison between different countries and the treatment accessibility. The new text now appears on page 33, lines 1095-1108, and reads as follows:
“Importantly, cost-effectiveness of targeted treatment approaches, exemplified by PARP inhibitors and PSMA-targeted radioligand therapy, presents a complex and often contentious landscape. For example, PSMA radioligand therapy meets Germany’s willingness-to-pay threshold at about €69,000 per quality-adjusted life year (QALY) but falls short in countries with stricter benchmarks based on models where only health-system costs are considered. When patient out-of-pocket expenses and productivity losses are added (“societal perspective”), even PSMA therapy often fails to demonstrate acceptable value in many high-income settings[137]. PARP inhibitors face even steeper economic hurdles, as their high cost result in incremental cost-effectiveness ratios above conventional cutoffs, approximately CAD$565,000/QALY in Canada and AU$144,000/QALY in Australia[138,139]. Within this context, treatment with a PARP inhibitor such as olaparib for a year can cost between US$100,000 and US$150,000 in the United States and around €5,000 monthly (approximately €60,000 yearly) in Germany, underscoring the financial challenges associated with accessing these therapies.”
Reviewer 2 – Comment 5
Including recent or ongoing clinical trials linking omics stratification to drug response would add relevance.
Reviewer 2 – Response to Comment 5
To address this comment, section 3.2.1. about clinical trials has been revised to acquire a clear focus on this issue. In addition, an overview and a summary of this section are now provided in the discussion section on pages 31-32, lines 1031-1036, and it reads as follows:
“At the same time, multiple reports shed light on omics-based biomarkers that are applied as stratification means within running clinical trials (summarized in the second section of this article and Table 3). An exemplary case of an adaptive clinical trial where omics biomarkers are guiding treatment is the ProBio trial, where ctDNA bi-omarker-based screening is used to randomize patients with advanced PCa into four treatment arms, with the possibility to adapt therapy based on the biomarker status[92].
Reviewer 2 – Comment 6
A brief section on anticipated future developments as AI-enhanced analysis would also help contextualize the evolution of precision oncology.
Reviewer 2 – Response to Comment 6
To address this comment, the following section about the use of AI-trained models for the future of precision oncology has been added on pages 33-34 and lines 1125-1139, and it reads as follows:
“Besides the advancements at the molecular level, AI-based tools are also increasingly used: First, multimodal large‑language‑models(LLM) that can incorporate a variety of modalities, including across pathology slides, imaging, genomics, and by guidelines, now achieve up to 80% decision accuracy in simulated tumor‑board cases, vastly outperforming standalone LLMs[142]. Second, early studies within collaborative European cancer research network such as EUCAIM (EUropean Federation for CAncer Images), OPTIMA(Optimal Treatment for Early-stage Prostate Cancer) and other networks suggest that federated models(machine‑learning systems trained across multiple sites, pooling the raw data in one place) match or modestly exceed public, centrally trained models; recent demonstrations involve >60 institutions, but further large-scale validation is still in progress[143]. Third, AI‑driven feature extraction radiomics, pathomics, and longitudinal ctDNA‐omics already predict immunotherapy response with up to 89 % (AUC 0.98–1.00) accuracy and are being embedded in adaptive trial designs[144]. Moving forward, complementing existing therapeutic approaches with the above multi-parametric models could lead to true precision oncology and improved treatment of advanced PCa. “
Reviewer 2 – Minor Comments
- Ensure consistent terminology (e.g., switch between “castration-resistant prostate cancer” and “CRPC”).
- Recheck some grammatical constructions (e.g., long and passive sentences could be simplified for clarity).
- Please cite the following papers about mCRPC and PSMA PET:
DOI: 1007/s00259-025-07133-1
DOI: 10.1002/advs.202305724
DOI: 3390/cancers15194809
Reviewer 2 – Response to Minor Comments
We thank the Reviewer for the valuable suggestions.
- Consistent terminology has now been adopted throughout the article
- We tried to simplify long and passive statements.
- These articles have helped us improve the scope of the paper.
- The suggested papers have been added to the manuscript. DOI: 1007/s00259-025-07133-1
This article has been updated in the review paper under the section on the PSMA therapy in the paper on page 14, lines 450-461, and below is the pasted text directly from the manuscript:
“The clinical detectability and quantification of PSMA expressions in PCa lesions are directly influenced by the choice of radiotracer and imaging technology. Although [^68Ga]-labeled agents have traditionally been the clinical standard, next-generation fluorinated compounds like [^18F]DCFPyL offer advantages, including improved image resolution and delayed imaging protocols (referring to acquisition of image scans at 90–120 minutes, or even up to 3 hours post-injection, rather than the standard image ac-quisition within an hour), which have demonstrated superior sensitivity for local and nodal staging in high- and very high-risk patients. For example, a recent study demon-strated that [^18F]DCFPyL PET/CT significantly improves the detection of loco-regional disease in patients with advanced PCa who are candidates for radical therapy, ensuring more accurate risk stratification, surgical planning, and more effective prioritization for PSMA-targeted therapies[78].”
DOI: 10.1038/s41391-022-00623-5 and DOI: 3390/cancers15194809
We would like to thank you for your recommendation to incorporate the studies by Ferriero et al. (Cancers 2023, 15(19), 4809) and Ferriero et al. (Prostate Cancer Prostatic Dis. 2024;27:89–94). We truly appreciate the importance of covering advances in mCRPC treatment as highlighted in these works.
However, the principal aim and focus of our review was on omics-based guidance in therapy selection. In our systematic search of Web of Science, one criterion was the exclusion of all studies where omics intervention was not involved. Our manuscript is specifically structured to summarize and critically appraise how high-throughput omics technologies (such as genomics, transcriptomics, proteomics, and related multi-omics platforms) are currently guiding, stratifying, or informing clinical therapeutic decisions and drug development in advanced prostate cancer.
After careful re-examination of the above-referenced studies, we noted that while they provide valuable real-world evidence on therapy sequencing and the impact of locoregional treatments in mCRPC, omics-based analyses or biomarker-driven patient stratification were not included in their study designs, methodologies, or interpretations. The studies are based solely on clinical, pathological, and imaging parameters, without integration of genomic, transcriptomic, or related molecular profiling data. As such, they were excluded based on our systematic search criteria. To preserve the clarity and thematic integrity of our review, we would respectfully opt not to discuss in the specific manuscript.
Round 2
Reviewer 2 Report
Comments and Suggestions for Authors
No more comments